# *Mycobacterium tuberculosis* requires SufT for Fe-S cluster maturation, metabolism, and survival *in vivo*

**Ashutosh Tripathi**[1], **Kushi Anand**[1], **Mayashree Das**[1], **Ruchika Annie O'Niel**[2], **Sabarinath P. S**[2], **Chandrani Thakur**[3], **Raghunatha Reddy R. L.**[4], **Raju S. Rajmani**[1], **Nagasuma Chandra**[3], **Sunil Laxman**[2], **Amit Singh**[1] *

1 Centre for Infectious Disease Research (CIDR), Department of Microbiology and Cell Biology, Indian Institute of Science (IISc), Bengaluru, India, 2 Institute for Stem Cell Science and Regenerative Medicine (inStem), Bangalore, India, 3 Department of Biochemistry, Indian Institute of Science, Bangalore, India, 4 Regional Horticultural Research and Extension Centre (RHREK), GKVK, Bengaluru, India

* asingh@iisc.ac.in

**Data Availability Statement:** All relevant data are within the manuscript and its Supporting Information files.

## Abstract

Iron-sulfur (Fe-S) cluster proteins carry out essential cellular functions in diverse organisms, including the human pathogen *Mycobacterium tuberculosis* (*Mtb*). The mechanisms underlying Fe-S cluster biogenesis are poorly defined in *Mtb*. Here, we show that *Mtb* SufT (Rv1466), a DUF59 domain-containing essential protein, is required for the Fe-S cluster maturation. *Mtb* SufT homodimerizes and interacts with Fe-S cluster biogenesis proteins; SufS and SufU. SufT also interacts with the 4Fe-4S cluster containing proteins; aconitase and SufR. Importantly, a hyperactive cysteine in the DUF59 domain mediates interaction of SufT with SufS, SufU, aconitase, and SufR. We efficiently repressed the expression of SufT to generate a SufT knock-down strain in *Mtb* (SufT-KD) using CRISPR interference. Depleting SufT reduces aconitase's enzymatic activity under standard growth conditions and in response to oxidative stress and iron limitation. The SufT-KD strain exhibited defective growth and an altered pool of tricarboxylic acid cycle intermediates, amino acids, and sulfur metabolites. Using Seahorse Extracellular Flux analyzer, we demonstrated that SufT depletion diminishes glycolytic rate and oxidative phosphorylation in *Mtb*. The SufT-KD strain showed defective survival upon exposure to oxidative stress and nitric oxide. Lastly, SufT depletion reduced the survival of *Mtb* in macrophages and attenuated the ability of *Mtb* to persist in mice. Altogether, SufT assists in Fe-S cluster maturation and couples this process to bioenergetics of *Mtb* for survival under low and high demand for Fe-S clusters.

## Author summary

*Mycobacterium tuberculosis* (*Mtb*- causative agent of tuberculosis) exploits Fe-S cluster containing proteins for respiration, metabolism, DNA repair, antibiotic resistance, and persistence. Therefore, the mechanism(s) underlying the biosynthesis of Fe-S clusters is important to understand the physiology of this human pathogen. Recent studies indicate

**Funding:** The Mtb work was supported by the following Wellcome Trust/DBT India Alliance Grants, IA/S/16/2/502700 (AS), and in part by Department of Biotechnology (DBT) Grant (BT/PR13522/COE/34/27/2015,BT/PR29098/Med/29/1324/2018, BT/HRD/NBA/39/07/2018-19, and BT/PR39308/DRUG/134/86/2021) (AS), DBT-IISc Partnership Program (22-0905-0006-05-987-436), DST-FIST grant, and the Revati and Satya Nadham Atluri Chair Professorship (AS). The funders had no role in study design, data collection and analysis, decision to publish, or preparation of the manuscript.

**Competing interests:** The authors have declared that no competing interests exist.

that proteins containing DUF59 domains participate in Fe-S cluster assembly in few organisms. However, the function of proteins-containing DUF59 domains is unknown in *Mtb*. We show that *Mtb* expresses a DUF59 containing protein SufT that functions as an auxiliary factor utilized in Fe-S cluster maturation. SufT physically interacts with other accessory proteins (SufS and SufU), coordinating Fe-S cluster biogenesis in *Mtb*. We also show that SufT is required to maintain the activity of Fe-S cluster proteins during normal growth conditions and under environmental settings that enforce a high demand for Fe-S clusters. Lastly, deficiency of SufT adversely affected *Mtb's* respiration, metabolism, growth inside macrophages, and ability to cause infection in mice.

## Introduction

The production of reactive- oxygen (ROI) and -nitrogen intermediates (RNI) by phagocytes is fundamental to resist microbial infections. Although ROI and RNI damage microbial DNA, lipids, and proteins, Fe-S clusters are exceptionally susceptible cellular cofactors [1, 2]. Since Fe-S cluster proteins are involved in multiple cellular processes (*e.g.*, respiration, central metabolism, DNA repair, gene regulation and RNA modification), microbes have evolved sensitive strategies to repair damaged Fe-S clusters for survival [3]. *Mycobacterium tuberculosis* (*Mtb*) (causative agent of tuberculosis [TB]) alleviates ROI and RNI in parts through Fe-S cluster proteins involved in redox-sensing, gene regulation, and DNA repair for persistence [4–10]. While these findings underscore the importance of Fe-S cluster proteins in ensuring stress tolerance and survival of *Mtb*, very little is known about Fe-S cluster biogenesis in this human pathogen. Filling this knowledge gap is critical to understand *Mtb's* survival strategies and to develop new therapeutic strategies.

The *Mtb* genome encodes an atypical Suf system (Rv1460-Rv1466; *sufRBDCSUT*) as the only complete Fe-S cluster biogenesis machinery [11]. Iron (Fe) is vital for the survival and persistence of *Mtb* [12]. Sequestration of Fe is a major component of nutritional immunity that protects from *Mtb* infection in humans [13]. However, studies suggest that *Mtb* successfully competes, acquires, and transports Fe from the host cells for persistence [12]. The majority of intracellular Fe (~80%) is assembled in the form of Fe-S clusters and heme in respiring microbes [14]. These findings indicate that the growth and survival of *Mtb* is likely to rely on the proper assembly of Fe-S clusters. Inadequate assembly of Fe-S clusters can result in metabolic paralysis and cell death [2]. The *suf* operon is the only Fe-S cluster biogenesis system present in *Mtb* [11]. All *suf* genes except Rv1460, which encodes a Fe-S cluster containing transcription factor (*sufR*)[15, 16] are predicted to be essential for viability [17]. The *suf* genes were upregulated under Fe-limitation, nitrosative stress, oxidative stress, and antibiotics [18–21], underscoring the importance of Fe-S biogenesis in *Mtb*. Consistent with this, lack of SufR compromised *Mtb's* ability to tolerate antibiotics, redox-stress, Fe- limitation, and survival in animals [16, 22].

Bioinformatic analysis predicted the function of *suf* genes in forming the Fe-S scaffold (Rv1461-Rv1463 [*sufBCD*] and Rv1465 [*sufU*]) and catalyzing S-transfer reactions from L-cysteine to S-acceptor molecules (cysteine desulfurase; Rv1464 [*sufS*]). The last gene of the *suf* operon (Rv1466 [*sufT*]) has no homology with any of the *suf* genes, except that it contains a DUF59 domain [23]. A truncated version of *Mtb* SufT partially complemented the growth phenotypes of the *Staphylococcus aureus* strain lacking a DUF59 containing SufT [24]. The *S. aureus* SufT is essential for maturation of Fe-S proteins such as aconitase (Acn), isopropylmalateisomerase (LeuCD), and dihydroxy-acid dehydratase (IlvD) under conditions of

heightened requirement for Fe-S cluster biogenesis [24]. *S. aureus* SufT is dispensable for Fe-S maturation under standard growth conditions or repair of damaged Fe-S clusters [24]. Similar to this, an initial report on SufT of *Mycobacterium smegmatis* (*Msm*- non-pathogenic and fast grower) was dispensable under standard growth conditions, but essential during Fe-limitation [25]. In contrast, *Mtb* SufT is predicted to be essential for viability under standard growth conditions [17], suggesting that the function of SufT is distinct in *Mtb* as compared to *S. aureus* or *Msm*.

In this study, we performed an *in silico* examination of *Mtb* SufT and identified the role of a conserved cysteine residue in Fe-S cluster maturation. SufT participated in Fe-S cluster maturation and repair by physically associating with other Suf proteins and Fe-S cluster acceptor proteins in *Mtb*. We systematically depleted SufT and examined bacillary growth under ambient conditions, upon exposure to redox stress, and inside macrophages and mice. Lastly, we assessed the effect of SufT depletion on TCA cycle metabolites, amino acid metabolism, and bioenergetics of *Mtb*. Our findings provide functional insights on how SufT contributes to Fe-S cluster maturation and cellular homeostasis to promote survival of *Mtb in vitro* and *in vivo*.

## Results

### *In silico* analysis indicates the role of SufT in Fe-S cluster maturation

The function of a large majority of DUF59-containing proteins remains unknown [26] except for a few DUF59 proteins involved in phenylacetic acid degradation and the maturation of iron-sulfur (Fe-S) cluster-containing proteins [24, 27–29]. The DUF59 domain-containing proteins display 9 modular structures (S1 to S9 modules in S1A Fig). The *Mtb* SufT DUF59 domain contains an extra N-terminal motif (S2 module in S1A Fig) that did not exhibit any homology to previously defined domains (S1A and S1B Fig). The DUF59 domain contains a conserved motif containing a hyperactive cysteine with heightened nucleophilicity (DPE-X26-31-T-X2/3-C) [30]. Both of these features are preserved in *Mtb* SufT (S1B Fig). The *Mtb* SufT NMR structure (PDB ID: 5IRD) shows a α/β-topology with five helices, and three-stranded, mixed parallel/anti-parallel β-sheets (Fig 1). The sequence arrangement of the regular secondary structures is α1–α2–β1–β2–α3–β3–α4- α5. We mapped the NMR structure of *Mtb* SufT on to the structure of proteins containing DUF59 domain in *Bacillus subtilis* YitW, (PDB ID: 3LNO), *Thermotoga maritima* TM0487 (PDB ID: 1UWD), and *Homo sapiens* FAM96a (PDB ID: 2M5H), and confirmed that the conserved D, E, T, and C residues are located commonly on solvent-exposed loops (Fig 1). However, while DET-associated hydroxyl side chains position towards one another to form a pocket with an average distance of ~ 5Å between them in SufT of *B. anthracis*, *T. maritima*, and *H. sapiens*, the same pocket in *Mtb* SufT exhibits an average distance of 12-17Å (Fig 1). Similar to *B. anthracis* and *T. maritima* DUF59, but not the *H. sapiens* DUF59, the cysteine thiol (C62) clusters with the DET-associated hydroxyl group in *Mtb* SufT (Fig 1). Moreover, the strict conservation of proline in position 63 is likely to have a role in properly arranging C62 in the active site of *Mtb* SufT (Fig 1). The overall arrangement of D, E, T, C, and P residues can therefore create an environment to bind a Fe ion or a Fe-S cluster (Fig 1) [31], although, a well-recognized Fe-S cluster binding motif could not be identified. The structural feature of *Mtb* SufT, along with its organization within the *suf* operon, suggests that SufT might have a role in Fe-S cluster maturation.

### *Mtb* SufT interacts with Fe-S cluster biogenesis proteins; SufS and SufU

Our *in silico* analysis suggests the importance of C62 and D, E, and T residues in creating a pocket for coordinating Fe or a Fe-S cluster in SufT. To examine this possibility, we purified histidine-tagged wild type (WT) SufT, or alanine mutant of the hyperactive cysteine residue

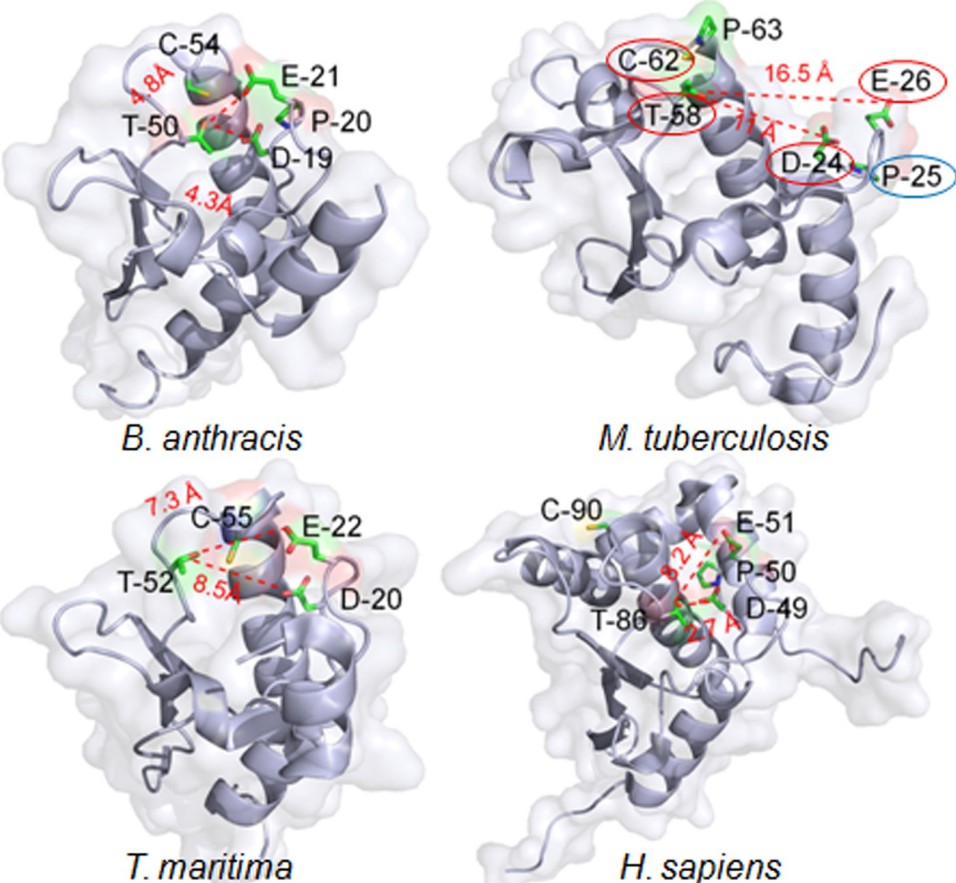

**Fig 1. Three-dimensional structure of Rv1466 (SufT).** *Mtb* SufT like other DUF59 proteins contains a small surface of conserved homology. Ribbon diagrams are shown for the *Bacillus anthracis* YitW (PDB ID: 3LNO); *Mtb* SufT (PDB ID: 5IRD); *Thermotoga maritima* TM0487 (PDB ID: 1UWD); and *Homo sapiens* FAM96a (PDB ID: 2M5H). The side chains of the conserved D, E, T, and C (red encircled) residues are highlighted. Oxygen, sulfur, and nitrogen atoms are represented with red, yellow, and blue color, respectively. Distance in Angstrom (Å) is represented between the side-chain hydroxyl of the T, D, and E residues. A proline (P25 in *Mtb*) present in the motif as DPE (blue encircled) and the consecutive proline (P63 in *Mtb*) next to the C residue are represented in the *Mtb* SufT structure.

(SufT$_{C62A}$) from *E. coli* cultured in growth conditions optimized for the maximum insertion of Fe or Fe-S clusters *in vivo* [16]. Purified SufT did not display the characteristic straw brown color associated with Fe or Fe-S cluster binding proteins. Agreeing to this, UV-visible spectroscopy did not reveal absorbance features associated with Fe (~ 420 nm) or Fe-S clusters (~ 340 nm [2Fe-2S], ~ 400 nm (3Fe-4S), ~ 420 nm [4Fe-4S]) (S2A Fig). Biochemical assays and atomic force spectroscopy confirmed the absence of Fe in SufT/SufT$_{C62A}$ or in bovine serum albumin (BSA; negative control), but presence in catalase (Fe-containing hemeprotein; positive control) (S2B and S2C Fig). Our results suggest that *Mtb* SufT is unlikely to bind Fe or a Fe-S cluster.

The DUF59 containing protein (YHR112W) in yeast mediates transfer of Fe-S clusters to acceptor/client proteins by physically interacting with the multiple proteins involved in Fe-S cluster assembly [30]. Importantly, the highly conserved cysteine residue in YHR112W likely regulates the formation of biologically active protein complexes with the Fe-S cluster biogenesis factors [30]. On this basis, we examined the interaction of SufT$_{WT}$ and SufT$_{C62A}$ with other Suf proteins involved in Fe-S cluster biogenesis in *Mtb*. Previous studies have shown that *Mtb*

SufT does not interact with other Suf proteins in the yeast-two-hybrid assays [11, 32]. While surprising, these studies are consistent in that certain *Mtb* protein: protein interactions require a natural host environment rather than a non-native host such as yeast [33]. Therefore, we used a mycobacterial two-hybrid system (mycobacterial protein fragment complementation assay [M-PFC]) to examine interacting partners of SufT [34]. The M-PFC system has been successfully used to demonstrate interactions between *Mtb* proteins using *Msm* as a surrogate. The M-PFC assay is based on the proximity-based reconstitution of murine dihydrofolate reductase (mDHFR) activity due to interaction between two proteins fused to mDHFR fragments F [1, 2] and F [3]. The reconstitution of mDHFR activity can be scored by detecting growth on a medium containing trimethoprim (TRIM), a drug that preferentially targets bacterial DHFR (Fig 2A). We tested the interaction of *Mtb* SufT with SufU and SufS, both of which are known to catalyze Fe-S cluster transfer to the client protein; the final step towards Fe-S cluster maturation. We co-expressed $SufT_{WT}$ or $SufT_{C62A}$ with SufU or SufS fused to the C-terminus of F [3] and F [1, 2] mDHFR fragments in *Msm* (designated as $SufT_{[F3]}$ or $SufT_{C62A[F3]}$ and $SufS_{[F1,2]}$ or $SufU_{[F1,2]}$), respectively. Additionally, the previously reported yeast GCN4 interacting pair ($GCN4_{[F3]}/GCN4_{[F1,2]}$) was taken as a positive control [34]. *Msm* co-expressing $SufT_{[F3]}/SufS_{[F1,2]}$, $SufT_{[F3]}/SufU_{[F1,2]}$, $SufT_{[F3]}/SufT_{[F1,2]}$, and $GCN4_{[F1,2]}/GCN4_{[F3]}$ combinations grew on 7H11 plates containing HYG, KAN and TRIM, indicating homodimerization of SufT and its interaction with SufS and SufU. In contrast, *Msm* co-expressing $SufT_{[F3]}/GCN4_{[F1,2]}$ did not show any growth on HYG/KAN/TRIM-containing medium. Interestingly, the substitution of Cys by alanine completely disrupted SufT homodimerization and association with SufS and SufU (Fig 2B). The expression of $SufT_{[F3]}$ and $SufT_{C62A[F3]}$ was comparable in *Msm* (S2D Fig), suggesting that mutation of the cysteine residue resulted in the loss of interaction.

## SufT physically associates with Fe-S cluster proteins aconitase and SufR

Having shown the interaction of SufT with SufS and SufU, we next asked if SufT directly associates with the proteins requiring Fe-S clusters for their function. To test this, we co-expressed SufT with 4Fe-4S cluster containing *Mtb* proteins Acn and SufR in *Msm* using the M-PFC vectors. We also co-expressed ESAT6 and CFP10 as these proteins associate strongly in mycobacteria [35]. The growth on HYG/KAN/TRIM containing medium was readily detected in the case of $SufT_{[F3]}/Acn_{[F1,2]}$, $SufT_{[F3]}/SufR_{[F1,2]}$, and $ESAT6_{[F1,2]}/CFP10_{[F3]}$, indicating interaction of SufT with Acn and SufR. *Msm* co-expressing $SufT_{C62A}$ with Acn or SufR did not grow on HYG/KAN/TRIM plates (Fig 3A).

We further validated M-PFC findings by re-examining the interaction between SufT and Acn in *Mtb* using *in vivo* pull-down assays. To do this, we utilized SufT-proficient (wt*Mtb* H37Rv) and a SufT-deficient strain of *Mtb*. Since *Mtb* SufT is proposed to be essential for the viability of *Mtb* [17], we generated a SufT-deficient strain using CRISPR-based RNA interference (CRISPRi)[36]. In CRISPRi, dCas9 and small guide RNA (sgRNA) expression was induced using anhydrotetracycline [ATc] from TetR-regulated promoter system. We first confirmed that 24 to 48 h exposure to 200 ng/mL of ATc resulted in 30–40 fold induction of dCas9 in vector control of *Mtb* (control) (S3A Fig). The expression of dCas9 alone in control strain of *Mtb* did not affect the growth under standard culture conditions (S3B Fig). Co-expression of dCas9 with the *sufT*-specific sgRNA in *Mtb* (SufT-KD) resulted in 10–60 fold down-regulation of *sufT* transcript at 24 h and 48 h post-exposure to 200 ng/mL of ATc (Fig 3B). Using an anti-SufT antibody, we confirmed an ~ 95% reduction in SufT levels at 48 h post-ATc treatment (Fig 3C).

We next performed immunoprecipitation of Acn from the cell-free lysate of the SufT-proficient control strain or the SufT-deficient (SufT-KD) strain. We observed comparable levels of

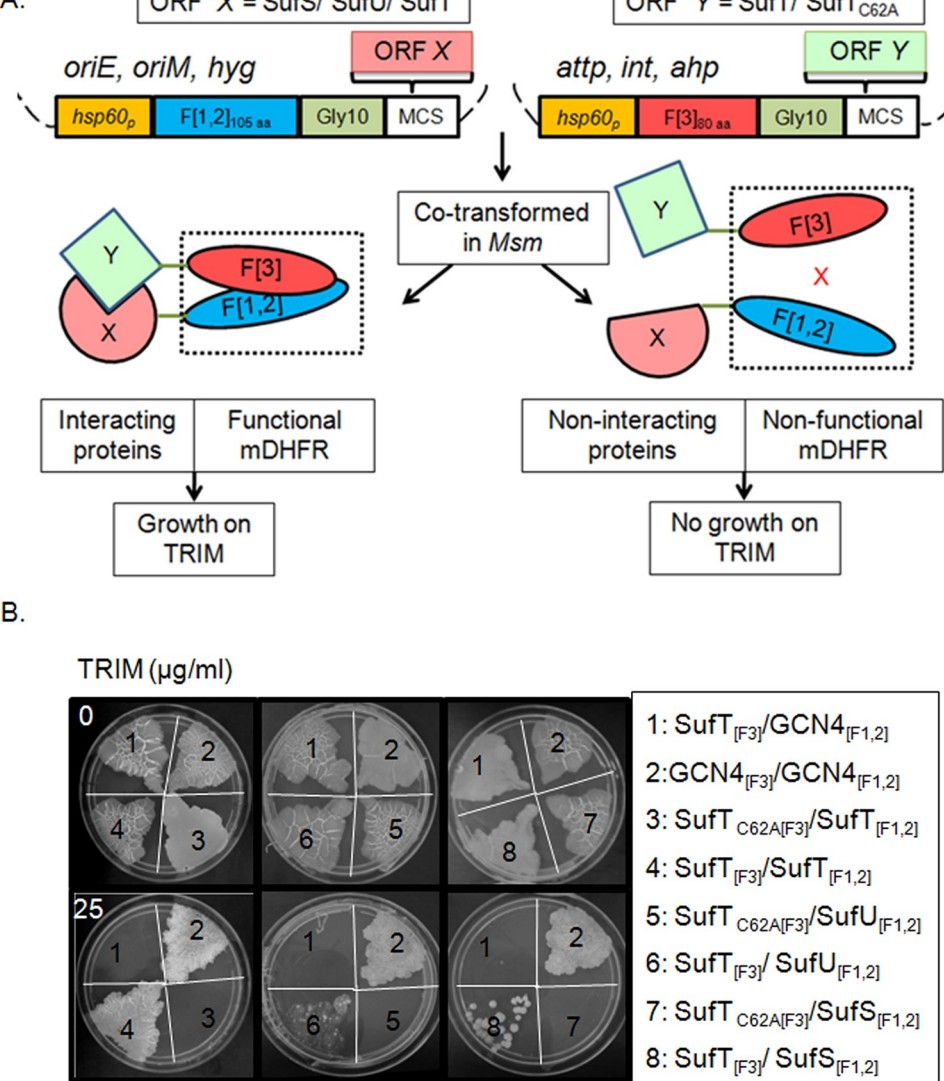

**Fig 2. SufT interacts with SufS and SufU of SUF system.** (A) Model illustrating the principle of M-PFC. Two independent plasmid constructs for candidate proteins X and Y are shown in fusion to complementary mDHFR fragments F [1, 2] and F [3], respectively. Co-transformation of XF [1, 2] and YF [3] fusions in *Msm* results in the functional reconstitution of mDHFR activity and subsequent growth on TRIM (0 and 25 μg/mL; concentration) plates, whereas proteins that do not interact will not reconstitute F [1, 2] and F [3] and consequently no growth on TRIM plates. Components of vectors, *aph* confers resistance to KAN; *hyg* confers resistance to HYG; *hsp60p* is the *hsp60* promoter; *oriM* is the origin of replication for propagation in mycobacteria; *oriE* is the origin of replication for propagation in *E. coli*; and *int* and *attP* are the integrase and phage attachment sites, respectively, from mycobacteriophage L5. (B) M-PFC based demonstration of C62 dependent specific protein-protein interaction of SufT with SufT, SufU and SufS. *Msm* as a host was co-transformed with plasmid construct expressing 1) SufT[F3]/GCN4[F1,2] (negative control), 2) GCN4[F1,2]/GCN4[F3] (positive control), 3) SufT[C62A[F3]]/SufT[F1,2], 4) SufT[F3]/SufT[F1,2], 5) SufT[C62A[F3]]/SufU[F1,2], 6) SufT[F3]/SufU[F1,2], 7) SufT[C62A[F3]]/SufS[F1,2] and 8) SufT[F3]/SufS[F1,2]. M-PFC experiments were repeated thrice, one representative image is shown here.

Acn in control and the SufT-KD strain (Fig 3D, upper panel). The proteins pulled down from the lysate using the anti-SufT antibody were separated and probed with the Anti-Acn antibody by immunoblotting. A strong signal was obtained in the case of the control strain, whereas the signal intensity was markedly reduced (~4 fold) in the SufT-KD strain (Fig 3D lower panel). A similar

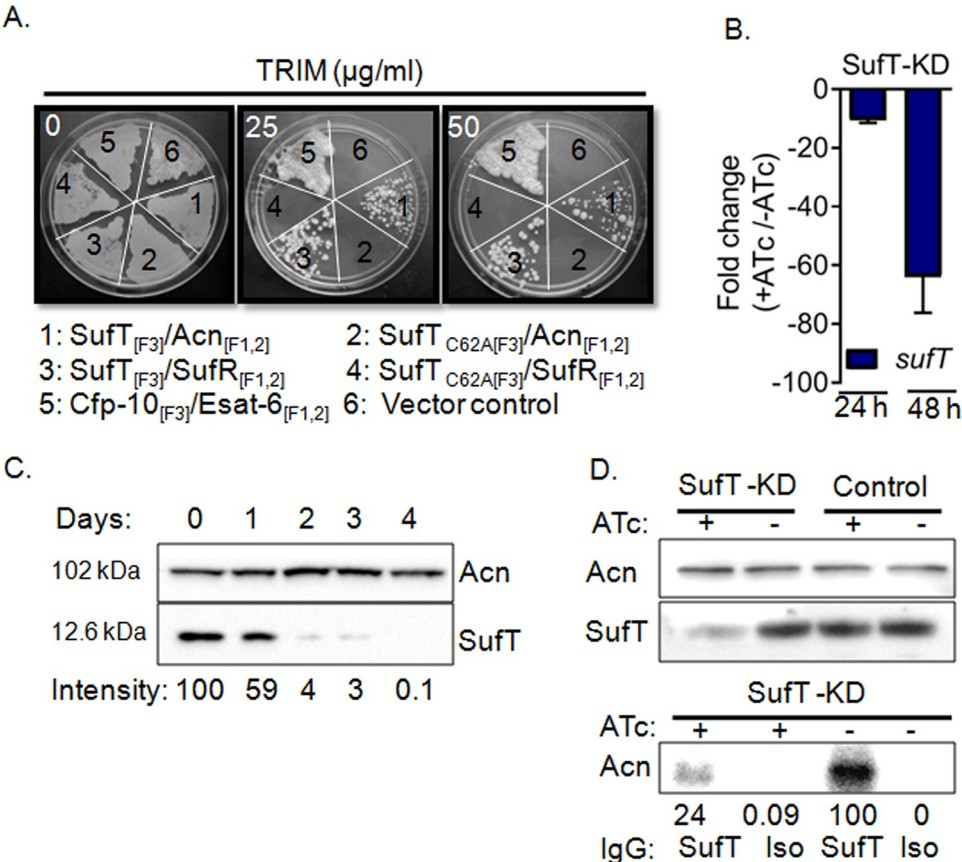

**Fig 3. SufT interacts with 4Fe-4S cluster containing proteins- SufR and Aconitase.** (A) M-PFC based demonstration of specific protein-protein interaction of SufT with Acn and SufR (4Fe-4S proteins), but not with $SufT_{C62A}$. Construct 1) $SufT_{[F3]}/Acn_{[F1,2]}$, 2) $SufT_{C62A[F3]}/Acn_{[F1,2]}$, 3) $SufT_{[F3]}/SufR_{[F1,2]}$, 4) $SufT_{C62A[F3]}/SufR_{[F1,2]}$, 5) $CFT10_{[F1,2]}/ESAT6_{[F3]}$ (positive control), and 6) empty vectors expressing [F3] and [F1,2] only (negative controls) were co-transformed in *Msm* and growth was scored on TRIM containing 7H11 plates. (B) The *Mtb* strain expressing pRH2502 (ATc inducible *dCas9*- vector control) and SufT specific gRNA in pRH2521 vectors (SufT-KD) was exposed to ATc (200 ng/mL) for 24 h and 48 h and the expression of *sufT* was monitored by RT-qPCR. (C) The SufT-KD strain was treated with 200 ng/mL of ATc and intracellular levels of SufT was determined at the indicated time points by immunoblotting using anti-SufT. Image density was analyzed with ImageJ software. Aconitase (Acn; 102 KDa) is used as loading control. (D) (top panel) Western blot of SufT and Acn from SufT-KD and vector control (with (+) and without (-) ATc) lysate. (lower panel) Immuno-precipitation of Acn from SufT KD (+ and − ATc) lysate with Anti-SufT IgG as well as rabbit Isotype IgG (Iso) as negative control. All M-PFC and western blot experiments were performed in triplicate, one representative image is shown here.

pull-down using a rabbit Isotype IgG antibody (IgG:Iso), yielded a low background signal of Acn (Fig 3D lower panel), affirming our findings that SufT specifically interacts with Acn in *Mtb*. In sum, our data suggest that SufT-mediated Fe-S cluster maturation requires direct interaction with proteins involved in Fe-S cluster biogenesis and Fe-S cluster requiring client proteins.

## SufT depletion reduces the activity of Fe-S cluster containing proteins

Next, we directly tested the role of *Mtb* SufT in Fe-S cluster maturation by measuring the activity of Acn, a known 4Fe-4S cluster-containing enzyme, in the SufT-KD strain. Acn activity was reduced to ~ 50% in the SufT-KD strain upon treatment with various concentrations of ATc for 72 h (Fig 4A). Under these conditions, the Acn levels remain unaltered; indicating that the loss of activity may be related to reduced 4Fe-4S cluster assembly of Acn (Fig 4A). We

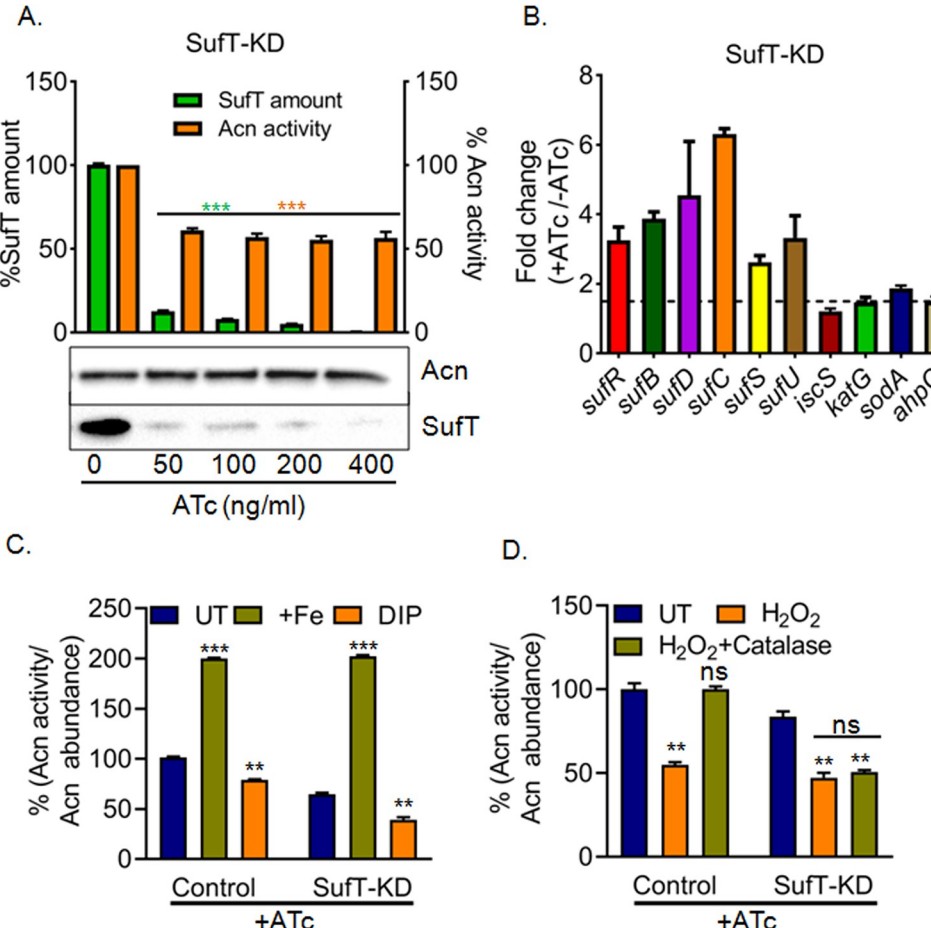

**Fig 4. SufT participates in Fe-S cluster biogenesis and repair.** (A) Western blot quantification of reduced SufT protein level (green bars) and decreased aconitase (Acn) activity (brown bars) was shown in SufT-KD strain, with increasing concentration (ng/mL) of ATc (0, 50, 100, 200 and 400). Level of SufT and enzymatic activity of Acn are normalized to Acn abundance from western blot. B) RT-qPCR based expression level of the *suf* operon (*sufR, sufB, sufD, sufC, sufS, sufU*), *iscS*, and anti-oxidant machinery (*katG, sodA and ahpC*) in SufT-KD strain. Horizontal dotted line indicates expression level below 1.5-fold considered as basal level. Data are shown as mean ±SD from triplicate reaction. (C) Acn activity was measured from cell free lysates derived from control and SufT-KD strains exposed to ATc (200 ng/mL) for 72 h and subjected to Fe-S reconstituted (Fe²⁺) and Dipyridyl (DIP) treatment. UT; untreated. D) Acn activity of UT, $H_2O_2$ and $H_2O_2$+Catalase treated cell lysates from control and SufT-KD strains were measured after 48 h of ATc (200 ng/mL) treatment. Student's t test (two tailed) was performed here with *p* values, $^*p = 0.05$, $^{**}p = 0.01$ and $^{***}p = 0.001$.

have recently shown that *Mtb* SufR, in a 4Fe-4S cluster bound form, represses the expression of *suf* operon [16]. Loss of 4Fe-4S cluster from SufR de-represses the *suf* operon [16]. Since SufT interacts with SufR, we expect that SufT is required for maturation of 4Fe-4S cluster on SufR to maintain its repressor activity. To examine this, we measured the expression of *suf* operon in the SufT-KD strain. As expected, the expression of *suf* operon was induced in the SufT-KD strain (Fig 4B), indicating that abnormal Fe-S cluster assembly on SufR upon SufT depletion likely resulted in the de-repression of the *suf* operon. *Mtb* encodes a homolog of cysteine desulfurase IscS (Rv3025c) that can insert the Fe-S cluster in *Mtb* proteins independent of the *suf* system [2, 37]. One possibility is that the depletion of SufT results in the reduced transcription of IscS system, which thereby diminishes the occupancy of the 4Fe-4S cluster on Acn and SufR. However, the transcription of *iscS* remained unaltered upon SufT depletion

(Fig 4B). Lastly, the 4Fe-4S clusters of Acn and SufR are sensitive to ROS [38, 39]. However, ROS in the SufT-KD strain remained comparable to the control strain (S4 Fig). Also, the expression of known antioxidant genes (*sodA*, *katG*, *ahpC*, and *ahpD*) was not affected in the SufT-KD strain (Fig 4B), clearly suggesting that defective Acn and SufR activities are not a consequence of Fe-S cluster damage due to heightened ROS. Based on the above results, we conclude that Acn and SufR activities are likely reduced due to decreased occupancy of the 4Fe-4S cluster upon SufT depletion.

## SufT repairs damaged Fe-S clusters in *Mtb*

Since respiration-linked ROS damages Fe-S clusters of Acn during aerobic growth [3, 24, 39], the steady-state activity of Acn reflects the equilibrium between the inactive (Fe-S cluster damaged) and active fraction of the enzyme [39]. The addition of ferrous ions (Fe[II]) shifts the equilibrium towards active fraction, whereas Fe(II) chelator dipyridyl had an opposite effect [40]. We asked if SufT is required to maintain the balance between the active and inactive fraction of Acn in *Mtb*. Acn activity was monitored in the cell-free extract of aerobically growing SufT-KD strain in the presence or absence of Fe(II). The addition of Fe(II) increased the activity of Acn in the control strain by 2-fold (Fig 4C), indicating that the enzyme is partially operational (50%) under steady-state conditions. Inclusion of Fe(II) increased the Acn activity by 3-fold in the SufT-KD strain, which amounted to Acn activity of only 30% upon under steady-state conditions (Fig 4C). Furthermore, the addition of 500 μM of dipyridyl (DIP) further decreased the steady-state activity of Acn by 22% and 40% in the control and the SufT-KD strains as compared to untreated cells, respectively (Fig 4C). This is consistent with an equilibrium shift towards inactive Acn fraction (lesser 4Fe-4S cluster) in response to SufT depletion under aerobic growing condition.

Transient exposure to $H_2O_2$ converts the active $[4Fe-4S]^{2+}$ form of Acn to inactive [3Fe-4S]$^{1+}$ form [39]. This can be repaired to the $[4Fe-4S]^{2+}$ active state by the cellular Fe-S cluster biogenesis machinery upon mitigation of $H_2O_2$ stress [24]. We exposed the cell-free lysate of 48 h ATc treated control and SufT-KD strains to 450 μM of $H_2O_2$ for 1 min and Acn activity was measured. In both cases, Acn activity was reduced by 50% in comparison to untreated cell lysates (Fig 4D). However, removal of $H_2O_2$ by catalase fully restored the Acn activity in the cell free lysates of control strain, but not in the lysate of SufT-KD strain (Fig 4D). Above findings suggest that SufT is required to maintain the steady-state activity of Acn and repair $H_2O_2$ damaged Fe-S clusters in *Mtb*.

## SufT depletion results in widespread metabolic realignment in *Mtb*

Since Fe-S cluster proteins are central to carbon metabolism, amino acid, and nucleotide biosynthesis [41], we assessed the influence of SufT depletion on these metabolites in *Mtb*. We utilized targeted, quantitative liquid chromatography-mass spectrometry (LC-MS/MS) based approaches (as described earlier [42] to analyze the steady-state amounts of amino acids, nucleotides and tricarboxylic acid cycle intermediates, in SufT-KD strain with and without ATc. Consistent with the reduced activity of Acn, the quantity of the Acn substrate (citrate) was elevated, and the product (α-ketoglutarate) was decreased in the SufT-KD strain (Fig 5A). Similarly, the substrate (fumarate) and the product (malate) of another 4Fe-4S cluster containing fumarase were increased and decreased, respectively (Fig 5A).

Accumulation of succinate indicates reduced activity of succinate dehydrogenase (SDH), which could be due to impaired maturation of Fe-S cluster on SDH in the SufT-KD strain (Fig 5A). A Fe-S cluster enzyme, lipoate synthase, provides lipoic acid to activate the E2 subunits of pyruvate dehydrogenase complex (PDHC) and H protein of the glycine cleavage system (GCS)

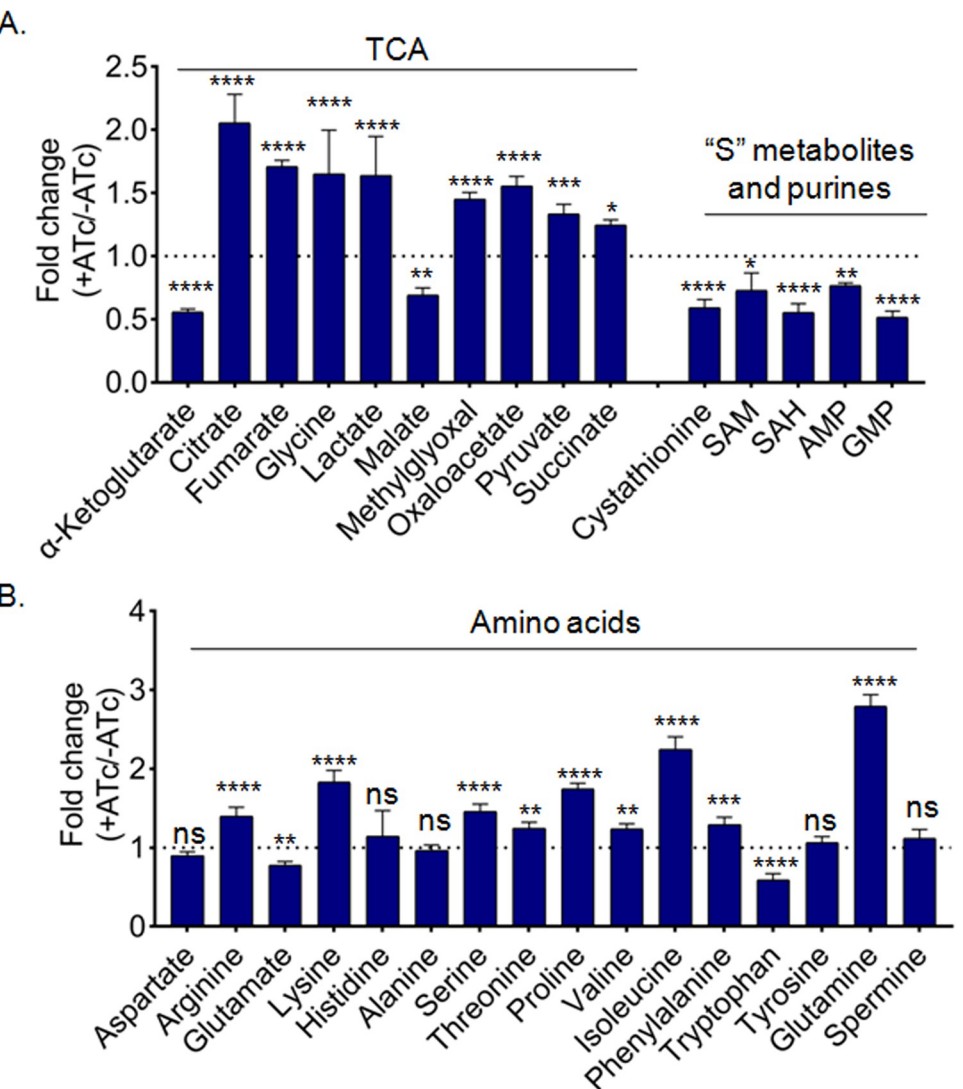

**Fig 5. Depletion of SufT results in deregulation of metabolites dependent on Fe-S cluster proteins.** Single column based LC MS/MS analysis of (A) OBHA derivatized TCA metabolites, underivatized measurement of purines, SAM, Cystathione (CTH), SAH and (B) Amino acids of SufT-KD strain post 72 h of ATc (200 ng/mL) treatment. Experiments were performed in biological triplicate and $p$ value was calculated by two-way ANOVA multiple comparison test. ns $p \geq 0.05$, *$p \leq 0.05$, **$p \leq 0.01$, ***$p \leq 0.001$.

[43]. Accumulation of glycine and pyruvate, which are the substrates of GCS and PDHC, respectively, indicates impaired lipoate synthase activity upon SufT depletion (Fig 5A). Similarly, valine and isoleucine that are catabolized by another lipoic-acid dependent enzyme (branched-chain keto-acid dehydrogenase complex [BCKDC])[44] were accumulated in the SufT-KD strain (Fig 5B). The depletion of TCA cycle intermediates, along with the abundance of some amino acids (*e.g.*, Arg, Lys, Ser, Thr, Prol, and Phe) (Fig 5B) suggest a compensatory mechanism to balance cataplerosis (removal of TCA cycle intermediates) by anaplerosis in the SufT-KD. Likewise, the sulfur-containing metabolites [*e.g.*, cystathionine (CTH), SAH, and SAM] were reduced, indicating disturbed sulfur metabolism in the SufT-KD. Due to relatively unstable nature of cysteine and methionine, we only measured CTH, SAM, and SAH as representative of sulfur amino acids. Further experiments are needed to fully understand the role of

SufT in sulfur metabolism. Lastly, we observed glutamine accumulation along with a decrease in the purine nucleotides in the SufT-KD strain (Fig 5A and 5B). This could be due to the reduced activity of a 4Fe-4S cluster enzyme glutamine phosphoribosylpyrophosphateamido-transferase (GPATase), which uses glutamine for de novo purine nucleotide biosynthesis [43, 45, 46]. Overall, data indicate that SufT is required for maintaining Fe-S homeostasis in *Mtb* in order to support carbon, amino acid, and nucleotide metabolism.

## SufT-depleted *Mtb* is bioenergetically deficient

A major realignment of central carbon metabolism upon SufT-depletion suggests altered bio-energetics. We utilized extracellular flux (XF) analyzer to noninvasively track the oxygen consumption rate (OCR) and extracellular acidification rate (ECAR) of the SufT-depleted strain. ECAR is an indication of proton ($H^+$) translocation linked to glycolysis and TCA cycle, while OCR is a reliable indicator of respiration due to oxidative phosphorylation (OXPHOS).

To quantify the basal and maximum rates of OCR and ECAR, we cultured *Mtb* in the XF microchamber, exposed to glucose, and then to the uncoupler carbonyl cyanide m-chlorophe-nyl hydrazine (CCCP) (Fig 6A). The addition of CCCP stimulates respiration to the maximal capacity manageable by *Mtb*. The difference between basal and CCCP induced OCR provides an estimate of the bioenergetics reserve, a capacity crucial for survival under redox stress [47]. Control and SufT-KD strains were grown in standard 7H9 media and treated with or without ATc for 72 h to deplete SufT. For measuring OCR and ECAR, cultures were diluted, and an equal number of SufT-KD and control cells ($2x10^6$ cells/well) adhered to XF cell culture plates as described in *Materials and Methods*. In the presence of glucose, the SufT-KD strain had a significantly lower OCR (57%) and ECAR (59%) than the control strain (Fig 6A and 6B). The addition of CCCP (5 μM) stimulated OCR and ECAR in both the strains however, the increase was significantly lower in the SufT-depleted strain (Fig 6A and 6B). The SufT-KD strain showed a 31% reduction in bioenergetics reserves (Fig 6C), exposing a likely vulnerability to redox stresses. By plotting the OCR and ECAR data, we generated a metabolic phenogram, which clearly showed that the SufT-depleted strain is metabolically derailed (Fig 6D). In sum, these data collectively implicate SufT in maintaining the bioenergetic homeostasis of *Mtb*.

## *Mtb* SufT is required for growth and promotes survival under oxidative stress, nitric oxide exposure, and inside macrophages

We first examined the phenotypic consequence of SufT depletion on growth of *Mtb* in liquid broth cultures exposed to various concentrations of ATc over time. As shown in (Figs 7A and S5A), increasing the concentration of ATc uniformly reduces the growth of the SufT-KD strain. Interestingly, on the solid agar medium, addition of 200 ng/mL of ATc nearly eliminated the growth of the SufT-KD colonies (Fig 7B). These results indicate the requirement of SufT for promoting growth under aerobic growing conditions.

Oxidative and nitrosative stress damage Fe-S clusters and increase the demand for Fe-S biogenesis and repair. Importantly, the *suf* operon showed heightened expression in response to hydrogen peroxide ($H_2O_2$) and nitric oxide (NO) exposure (19, 20). On this basis, we tested the requirement of SufT for adaptation of *Mtb* under $H_2O_2$ and NO stress. Exposure to 10 mM $H_2O_2$ resulted in ~ 3-fold ($p = 0.0017$) decreased survival of the SufT-KD strain relative to unstressed bacteria (Fig 7C). However, a significant effect of NO on bacterial survival (~ 2.5-fold) ($p = 0.0002$) was only evident at 1mM of DETA/NO (Fig 7D). We next assessed the survival of SufT-KD inside macrophages. We infected PMA-differentiated THP-1 macrophages and resting RAW264.7 murine macrophages with the control and SufT-KD strains at multiplicity of infection (MOI) 2 and monitored the survival over time. In these models,

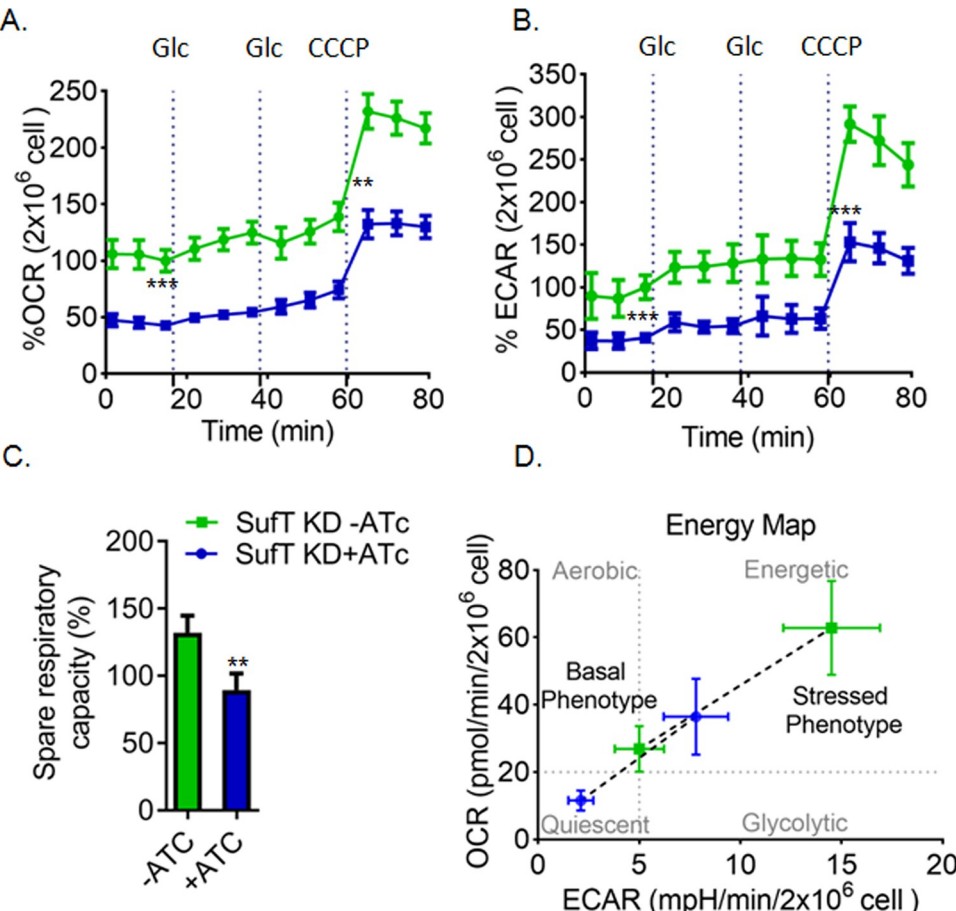

**Fig 6. Depletion of SufT results in reduced OCR and ECAR.** (A) A change in % oxygen consumption rate (% OCR) and (B) % extra cellular acidification rate (% ECAR) of SufT-KD strain precultured for 72 h in the presence (+ATc) or absence (–ATc) of ATc and was measured after injection of 5 mM glucose (Glc) and 5 μM CCCP indicated by dotted lines. The uncoupler CCCP was used to determine the spare respiratory capacity (SRC). (C) Decrease in 20% SRC of SufT-KD+ATc with respect to SufT-KD-ATc. Percentage SRC was calculated by subtracting basal OCR (point before adding first Glc) to CCCP induced OCR. (D) Agilent Seahorse XF based Cell Energy Phenotype Profile of SufT-KD. The relative utilization of the two energy pathways (oxidative phosphorylation and glycolysis) is determined under both baseline (Baseline Phenotype) and stressed (Stressed Phenotype) condition after CCCP addition. All point of OCR and ECAR are normalized to CFU ($2 \times 10^6$ cells/well) and calculated in percent with respect to third basal point. Seahorse XF analyzer experiment was performed in triplicate with two independent replicates. Student's t test (two tailed) was performed here with $p$ values, $^*p \leq 0.05$, $^{**}p \leq 0.01$ and $^{***}p \leq 0.001$.

control and SufT-KD strains without the ATc inducer showed unrestrictive growth over time (Fig 7E and 7F). In contrast, ATc pretreatment for 72 h, prevented the proliferation of SufT-KD in THP-1 and RAW264.7 macrophages. Lastly, we treated RAW264.7 with the iNOS inhibitor 1400W [48] and examined the growth phenotype of the SufT-KD strain. The growth inhibition exhibited by the SufT-KD strain was partially reversed upon treatment of RAW264.7 with 1400W (S5B Fig). Taken together, these data suggest that SufT provides resistance to oxidative and NO stress and is required for survival of *Mtb* inside macrophages.

## SufT is required for growth and persistence of *Mtb* in mice

We next sought to determine the impact of SufT depletion on the growth and persistence of *Mtb* in mice. The Tet-based regulatory system used in this study to deplete SufT has been

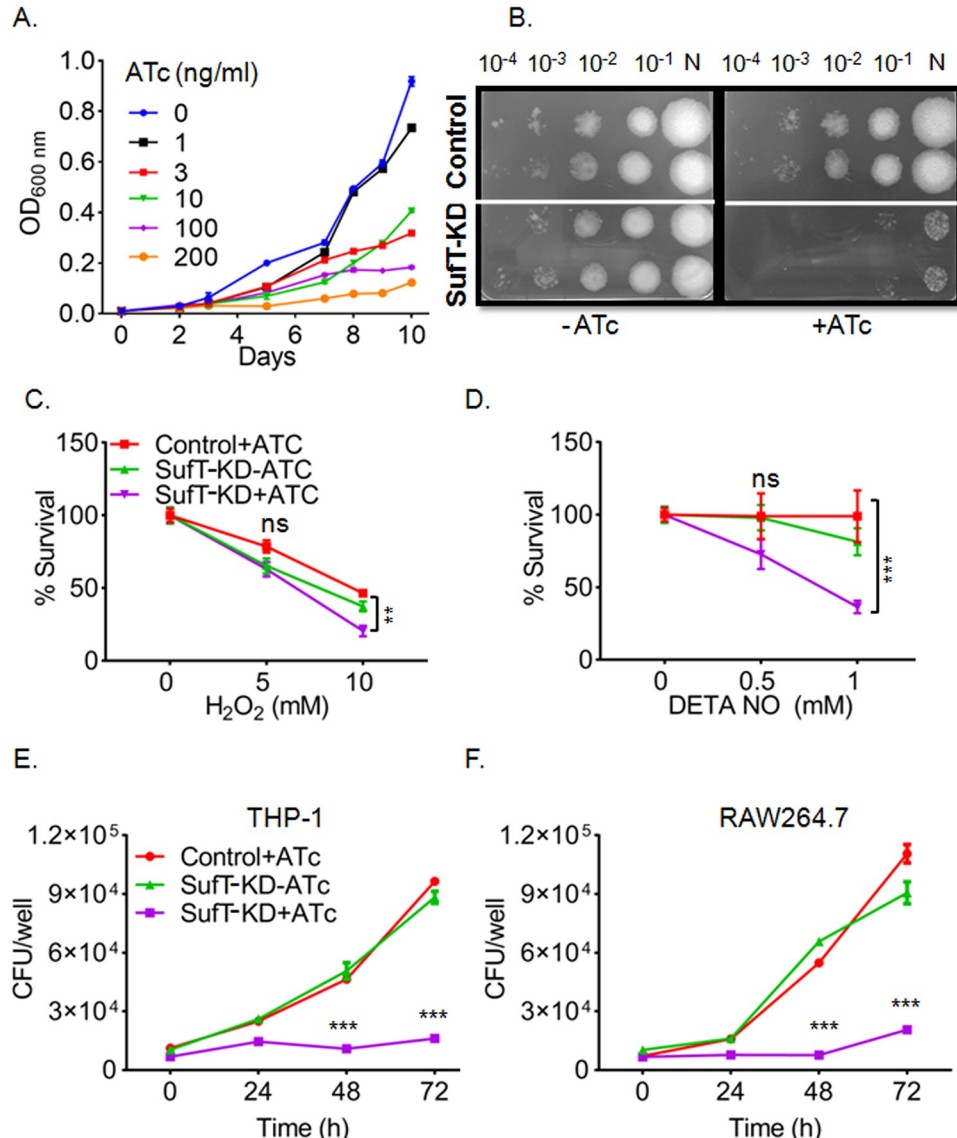

**Fig 7. SufT is essential for *in vitro* growth and regulates survival under redox stress and inside macrophages.** (A) A gradual decrease in the growth of the SufT-KD strain with increasing concentration of ATc (0–200 ng/mL) in standard liquid 7H9 medium. (B) Image of the serially diluted (N = undiluted) culture spotting of control as well as SufT-KD on 7H11 plates with and without ATc (200 ng/mL). Log phase culture of control +ATc, SufT-KD+ATc and–ATc were diluted to 0.15 OD (~2x $10^7$ cells/mL) and treated with (C) $H_2O_2$ (0 mM, 5 mM and 10 mM) or (D) DETA NO (0 mM, 0.5 mM and 1 mM) for 12 h. *Ex vivo* survival of control +ATc, SufT-KD +ATc, and–ATc was performed in (E) THP-1 and (F) RAW264.7 cell line. Infection was performed at MOI: 2 for 4 h (0 h). CFU was measured at 0, 24, 48 and 72 h of infection. Experiments were performed in triplicate with two independent experiments. Two-way analysis of variance (ANOVA) as applicable with multiple comparisons test was employed to determine statistical significance. $^{**}p \leq 0.01$ and $^{***}p \leq 0.001$.

successfully utilized to analyze the requirement of essential genes in *Mtb* during acute and chronic phase of infection [49]. We infected BALB/c mice with the SufT-KD strain by aerosol. As previously reported [49], we initiated SufT depletion at day 7 post-infection to assess acute phase phenotype and for the chronic phase at day 21 post-infection by feeding doxycycline (DOX), a tetracycline derivative frequently used for gene silencing *in vivo* [50]. Wt *Mtb* proliferated for 21 days in mice followed by the onset of adaptive immunity resulting in the

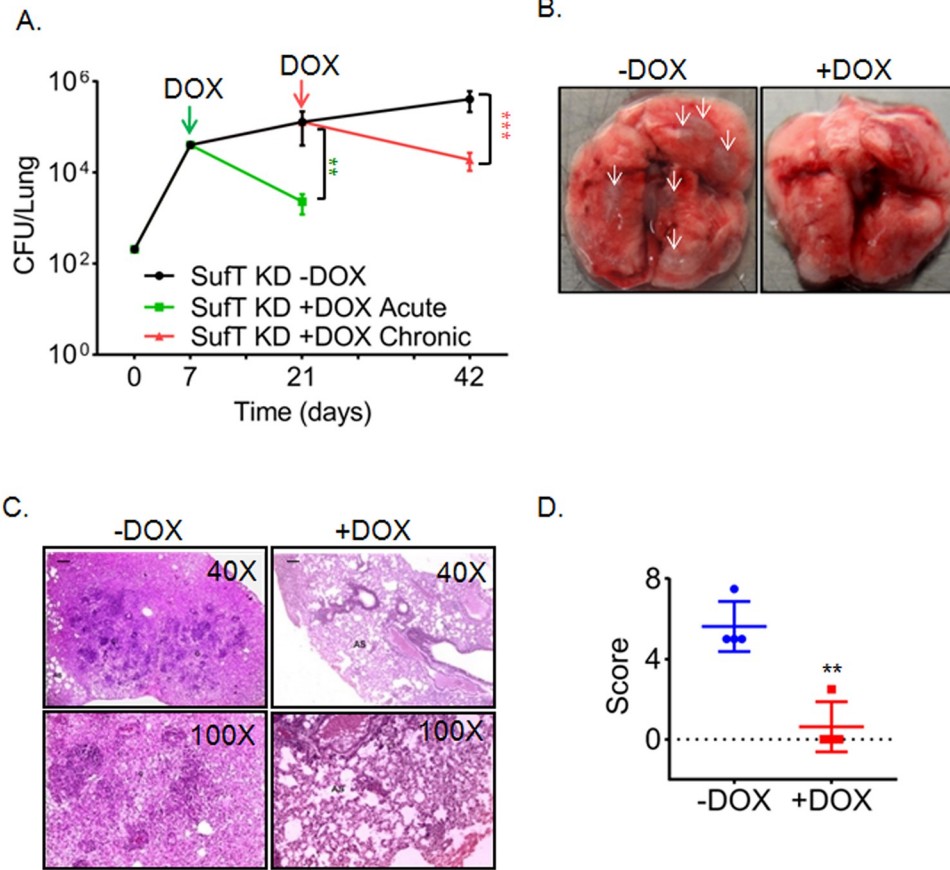

**Fig 8. SufT is required in acute and chronic phase of infection *in vivo*.** (A) BALB/c mice (n = 6) were aerosol infected with SufT-KD strain at 200 CFU per lung at 0 day and divided into three groups of i) SufT-KD -DOX (no doxycycline treatment), ii) SufT-KD +DOX Acute (DOX started at 7 days after infection) and iii) SufT-KD +DOX Chronic (DOX started at 21 days after infection). Post infection six animals were sacrificed from each group on day 7, 21 and 42 and CFU load per lung was measured. (B) Gross pathology of lungs (upper panel) from SufT-KD -DOX and +DOX infected BALB/c mice was shown after 42 days of infection. White arrows indicate distinct granulomas. (C) Hematoxyline and eosin- stained lung sections after 42 days of infection. (D) Histo-pathological granuloma scoring of lung sections from–DOX and + DOX infected BALB/c mice. Student's t test (two tailed) was performed to determine statistical significance as **$p \leq 0.01$ and ***$p \leq 0.001$.

stabilization of bacterial load in the lungs. The SufT-KD strain multiplied at a rate comparable to wt *Mtb* in the absence of DOX [49]. However, bacterial counts were reduced to 56-fold ($p = 0.0073$) at 21 days and 21-fold ($p = 0.00079$) at 42 days post-infection when SufT depletion was initiated during the acute (day 7) and chronic (day 21) stage, respectively (Fig 8A). The gross and histopathological changes in the lungs of the SufT-KD strain after 42 days of infection were comparable with the bacillary load observed (Fig 8B and 8C). The extent of pulmonary tissue damage was significantly greater in the SufT-KD strain in DOX untreated animals (score 5.6) relative to DOX treated animals (score 0.6) (Fig 8D). Since ATc/DOX addition sustains dCas9 expression only for a few days [36], we feel that a relatively smaller effect of SufT depletion on the survival and persistence of *Mtb* is likely due to decrease in dCas9 induction and SufT depletion over time. Altogether, these results confirm that SufT is necessary for *Mtb's* survival during all stages of infection in mice.

## Discussion

The presence of a DUF59 domain on *Mtb* SufT and its physical linkage with other *suf* genes strongly suggest a possible role in Fe-S cluster biogenesis. Here, we have presented exhaustive genetic, biochemical, and animal studies to show that DUF59 containing *Mtb* SufT physically interacts with Fe-S cluster biogenesis proteins and Fe-S cluster proteins to coordinate metabolism, bioenergetics, growth, and stress survival *in vitro* and *in vivo*. The physiological deficiencies manifested by the SufT-depleted strain are indicative of defects in Fe-S cluster maturation, as described for other DUF59 containing SufT homologs [24, 27, 51, 52]. For example, DUF59 domain-containing proteins from *Arabidopsis thaliana* (*e.g.*, HCF101, AE7), humans (Fam96a, Fam96c), *S. aureus* (SufT), and *Saccharomyces cerevisiae* (CIA2A and CIA2B) are involved in the maturation of Fe-S clusters to control central metabolism, respiration, photosynthesis, and growth.

However in contrast to the major requirement of other SufT homologs for *de novo* Fe-S cluster assembly upon apo-protein [24, 27, 29, 51, 52], *Mtb* SufT seems to also protect Fe-S clusters from leaching (*e.g.*, dipyridyl) and reactivation of $H_2O_2$-damaged Fe-S clusters. While the mechanism of SufT mediated protection, maturation, and regeneration of Fe-S clusters needs further experimentation, data showing a direct interaction between SufT, SufU and SufS suggest its function in the carriage and transfer of Fe-S clusters to the client proteins during assembly. In line with this, DUF59 containing *A. thaliana* HCF101 binds 4Fe-4S cluster and transfers the cluster to an acceptor apo-protein akin to an Fe-S cluster carrier [51]. However, *Mtb* SufT contains only one strictly conserved cysteine, whereas Fe-S carriers generally contain two or more cysteine residues for Fe-S cluster coordination [31]. Consistent with this, we found that *Mtb* SufT does not bind Fe or Fe-S cluster, rather the conserved cysteine residue is essential for establishing interaction of SufT with SufS, SufU, Acn, and SufR. A likely possibility is that the association of SufT with Acn and SufR might be required for the physical elimination of Fe-S cluster damaging agents (*e.g.*, $H_2O_2$, dipyridyl, NO) from the Acn and SufR active site.

In addition to the DUF59 domain-containing proteins, organisms encode multiple auxiliary factors for Fe-S cluster maturation under prevailing environmental conditions [53–56]. For example, both SufT and Nfu perform Fe-S maturation of Acn in *S. aureus* [24]. However, Nfu is preferred under normal growth conditions, whereas SufT prevails under stress conditions [24]. Homologs of accessory proteins performing Fe-S maturation have not been identified in *Mtb*, indicating a broader role for *Mtb* SufT in Fe-S cluster homeostasis under ambient and stress conditions. Consistent with this premise, the SufT-depleted strain not only displayed defective growth in standard culture media but also showed poor survival under highly oxidative or nitrosative conditions. In the absence of multiple auxiliary factors, *Mtb* likely responds to a graded demand for Fe-S cluster biogenesis by regulating the expression of *sufT* via the transcriptional regulator SufR [16]. Under standard growth conditions, SufR coordinates a 4Fe-4S cluster and maintains the basal expression of the *suf* operon [15]. Stress conditions (*e.g.*, NO, ROS) damage the SufR Fe-S cluster resulting in a loss of repressor function and an induction of the *suf* operon, including *sufT* [19].

The physiological role of SufT in Fe-S cluster maturation is supported by the metabolite analysis, which shows impaired pool of TCA cycle intermediates, and accumulation of substrates utilized by the lipoamide-requiring enzymes (PDH, BKC, GCS). The synthesis of lipoic acid requires *Mtb* LipA, which utilizes two [4Fe-4S] clusters for catalysis [57]. Our data suggest that multiple phenotypes associated with SufT-depletion could be a consequence of decreased activities of the lipoate-requiring enzymes. Importantly, *S. aureus* SufT contributes to Fe-S

maturation under conditions of high demand for lipoic acid [58]. Further experiments are needed to confirm if SufT is involved in the maturation of 4Fe-4S clusters of LipA in *Mtb*.

Cysteine is the source of sulfur required for the Fe-S cluster formation. *Mtb* preferentially utilizes methionine to generate cysteine via methionine cycle intermediates (SAM, SAH, homocysteine, and cystathionine) using the reverse transsulfuration pathway [59]. Since the accumulation of cysteine is toxic in *Mtb* [60], it is utilized to form mycothiol [61], and also consumed by cysteine desulfurases (IscS and SufS) to release sulfane sulfur for Fe-S biogenesis [37]. Since methionine and SAM are essential metabolites [62], they are regenerated from cysteine and homoserine via the transsulfuration and homoserine transacetylase pathways, respectively [59, 62]. Other than methionine, inorganic sulfate can be assimilated to produce cysteine by the action of a Fe-S cluster-dependent enzyme APS reductase (APSR) encoded by *cysH* [60]. The balance between these pathways likely ensures the balance of methylation, antioxidant, and Fe-S cluster homeostasis in *Mtb* [60, 62, 63]. Our data provide new insight into this important metabolic branch point. Expectedly, the levels of TCA cycle metabolites and amino acids linked to the activity of Fe-S cluster enzymes were altered upon SufT-depletion. However, an unexpected finding was the diminished amounts of SAM, SAH, and cystathionine in the SufT-depleted strain. This could well be a consequence of the unregulated induction of the *suf* operon upon SufT-depletion. Due to defective Fe-S maturation of the repressor SufR, the SufT-KD strain constitutively overexpresses a cysteine desulfurase (*sufS*), which release sulfur from cysteine for Fe-S biogenesis. This increase in *sufS* likely results in increased utilization of cysteine for repairing Fe-S clusters upon SufT-depletion. A consequence of this would be the net loss of SAM, SAH, and cystathionine, which also depend on the methionine and cysteine pool via the methionine cycle and transsulfuration pathways. Moreover, defective Fe-S maturation would have rendered the Fe-S cluster-dependent APSR pathway an unlikely route to generate cysteine, further raising the dependency of the SufT-KD strain on the reverse transsulfuration for cysteine formation. All of this could lead to an exhaustion of SAM, SAH, and cystathionine in the SufT-KD strain. Because methionine and SAM are critical for host infection and the virulence of *Mtb* [62], the malfunctioning of the SufT-KD strain *in vitro* and *in vivo* is likely due to the collapse of multiple mechanisms associated with Fe-S cluster and methionine metabolism. This observation is consistent with the fact that the activity of several Fe-S cluster dependent enzymes are also contingent on SAM (e.g., *lipA*, methyltransferase) [57, 64].

In summary, in this study, we used *in vivo* and *in vitro* approaches to define the role of DUF59 containing SufT in the physiology and virulence of *Mtb*. We used a SufT-depleted strain as a model system to examine the consequence of defective Fe-S metabolism on growth, stress response, metabolome, bioenergetics, and pathogenesis. In doing so, we discovered a previously unknown link between the Fe-S cluster and methionine metabolism in *Mtb*. The study presented provides a framework for future studies examining the link between Fe-S homeostasis and pathogenesis of human pathogens such as *Mtb*.

## Materials and methods

### Ethics statement

**Animal experimentation.** This study was carried out in strict accordance with the guidelines provided by the Committee for the Purpose of Control and Supervision on Experiments on Animals (CPCSEA), Government of India. The protocol of animal experiment was approved by animal ethical committee on the Ethics of Animal Experiments, Indian Institute of Science (IISc), Bangalore, India (Approval number: CAF/Ethics/544/2017). All efforts were made to minimize the suffering.

## Bacterial strains and culture conditions

The *Mycobacterium tuberculosis* H37Rv, and derived strains (control and SufT-KD) were grown in Middlebrook 7H9 broth (Becton, Dickinson and Company, USA) medium supplemented with 0.2% glycerol, 0.5% BSA, 0.2% dextrose, and 0.085% NaCl (ADS) with 0.05% Tween-80 as described previously [65]. ATc was added in secondary culture between $OD_{600nm}$ 0.1–0.2 at concentration of 200 ng/mL [36]. For culturing on solid medium, *Mtb* strains were cultured on 7H11 agar medium (Becton, Dickinson and Company, USA) supplemented with 1x OADC (Becton, Dickinson and Company, USA) and 0.2% glycerol. Whenever required, antibiotics were added to the culture medium, at concentration of 25 μg/mL kanamycin (KAN) (Amresco, USA) and 50 μg/mL hygromycin (HYG) (Sigma-Aldrich, USA). *E. coli* DH5α and BL21 strains were grown in LB medium (HIMEDIA, India) with antibiotic concentration of 50 μg/mL KAN and 100 μg/mL HYG.

## Strain and plasmid construction

For construction of SufT-KD strain, CRISPR interference (CRISPRi) technology was used as described [36]. An inactive version of *Streptococcus pyogenes cas9* having mutations D10A and H820A (*dcas9*) was expressed by pRH2502 from a TetR-regulated uvtetO promoter. Gene specific small guide RNA (sgRNA) was expressed in pRH2521 vector under the control of a TetR-regulated smyc promoter ($P_{myc1}tetO$). To deplete *sufT*, gene specific sgRNAs were designed for two regions between 98–127 bp and 171–191 bp of *sufT* ORF and cloned in pRH2521. The SufT-KD secondary liquid culture in 7H9 was divided equally when $OD_{600nm}$ reached to 0.1–0.2 and cultured in the presence or absence of ATc (200 ng/mL). Depletion of *sufT* was verified by RT-qPCR. Based on the significant repression of *sufT*, we chose sgRNA targeting sufT region between 98–127 bp for further study. All the experiments were performed with 72 h ATc treated SufT-KD culture, unless otherwise mentioned.

## Bioinformatic analysis

The DUF59 family protein sequence alignment of amino acid residues from the *H. sapiens*, *S. meliloti*, *S. aureus*, *B. anthracis*, *M. smegmatis*, *M. tuberculosis*, and *M. leprae* were performed with ClustalW and identity of Rv1466 with homologous sequence distribution was achieved by BLASTp. The three-dimensional structural model for *Mycobacterium tuberculosis* SufT (PDB ID: 5IRD), *Bacillus anthracis* YitW (PDB ID: 3LNO), *Thermotoga maritima* TM0487 (PDB ID: 1UWD), and *Homo sapiens* FAM96a (PDB ID: 2M5H) were downloaded from RCSB PDB, a protein structure database (*www.rcsb.org*)[66]. The structures were visualized and analyzed in Pymol [The PyMOL Molecular Graphics System, Version 2.0 Schrödinger, LLC]. The DPE TC motif residues were highlighted and the distances between the selected residues were calculated.

## RNA isolation and RT-qPCR experiments

Total RNA extraction was conducted using the FastRNA Pro Blue Kit (MP Biomedicals, USA) in accordance with the manufacturer's instruction and further purified using RNeasy spin columns (Qiagen, USA) as described in [67]. For RT-qPCR analysis, total RNA was extracted from SufT-KD and control strains after 24 h and 48 h of ATc treatment. After DNase treatment, total 600 ng of RNA was used for cDNA synthesis by using random hexamer oligonucleotide primer (iScript Select cDNA Synthesis Kit, BioRad, USA). Gene specific primers (S1 Table) and iQ SYBER Green Supermix (BioRad, USA) were used for RT-qPCR (StepOne Plus, Thermo, USA). In order to obtain meticulous expression levels, PCR expression was

normalized with *Mtb* 16S rRNA expression as endogenous control and StepOne software v2.3 (Thermo) was utilized for data analysis.

## Aconitase assay

The Acn activity was measured by monitoring the disappearance of *cis*-aconitate at wavelength (λ) 240 nm in a UV spectrophotometer (Thermo Scientific Biomat 3S, USA) as described [68–70]. One unit (U) of Acn activity is defined as 1 μmol *cis*-aconitate formed or converted per minute. Reaction mixtures for Acn assay contained 25 mM Tris-HCl (pH 8.0), 100 mM NaCl, and 50 μg *Mtb* cell lysates in 1 mL of reaction volume. Reaction was initiated by adding 0.15 mM *cis*-aconitate and monitored by following the disappearance of *cis*-aconitate at λ 240 nm after every 15 sec for 30 min [68, 69]. Absorbance at λ 240 nm was plotted against time. Acn activity was calculated from linear portion of the curve in initial 5 min. An extinction coefficient of 3,500 $M^{-1}cm^{-1}$ was used to calculate the rates.

## ROS measurement

Exponentially growing SufT-KD culture at an initial $OD_{600\,nm}$ of 0.15 was divided in two and was either treated with 200 ng/mL of ATc for 72 h or left untreated. The cells were harvested by centrifugation at 4500x*g* for 5 min and re-suspended in 200 μL of growth medium. As per manufacturer instructions, CellRox Deep Red (Invitrogen, Waltham, MA) was added to a final concentration of 5 μg/mL and cells were agitated on a rocker (Biobee Tech, Bangalore, India) for 30 min. After incubation, cells were washed to remove residual dye by centrifugation at 4500x*g* for 5 min. Cells were re-suspended in 300 μL phosphate-buffered saline (PBS), (pH 7.4) and then fixed by addition of 4% paraformaldehyde (PFA) for 1 h at room temperature. Fluorescence was measured at a fixed emission (670 nm) after excitation with a red laser (640 nm) using a BD FACSVerse Flow cytometer (BD Biosciences, San Jose, CA).

## Western blot and immune-precipitation

For western blot, whole cells lysate (30 μg) was separated on 12% SDS-PAGE and then transferred onto a PVDF membrane (GE Healthcare, Piscataway, NJ, USA). Membrane was blocked using 5% (w/v) non-fat dry milk and incubated for 3 h at room temperature with primary antibody (Acn and SufT at 1:10000 dilutions). After washing with 1xTBST (Tris-buffered saline pH 7.5 with 0.05% Tween-20), membranes were incubated in goat anti-rabbit IgG HRP-conjugated secondary antibody (1:10000 dilution) for 1 h. The autoradiography signals were visualized using ECL advance Western blotting detection kit (BioRad, USA).

Immunoprecipitation of Acn and SufT was performed by using protein A/G Mix magnetic bead (Cat. No.- LSKMAGAG02 Sigma Aldrich, USA) as per manufacturer's protocol. Equal amounts of cell free lysates (200 μg lysate of SufT-KD strain with and without ATc) were incubated with anti–SufT IgG antibody at 4˚C for 1 h. Further 25 μL of protein A/G Mix magnetic beads were added and incubated with the antibody-lysates mix at 4˚C for 4 h with continuous mixing. The beads were washed thrice with PBST (PBS with 0.01% Tween-20) and eluted with acidic glycine (pH 2.5) or boiled with gel loading 1X laemmli buffer directly as per requirement of experiments. The eluates were separated on 12% SDS-PAGE gel and probed with anti-Acn IgG antibody.

## Mycobacterial protein fragment complementation (M-PFC) assay

The M-PFC assay is based on the functional reconstitution of murine dihydrofolate reductase (mDHFR) enzyme activity mediated by interaction between two candidate proteins fused to

mDHFR fragments F [1, 2] and F [3]. The reconstitution of mDHFR activity can be scored by detecting growth on 7H11 plate containing trimethoprim (TRIM) 0, 25 and 50 μg/mL, a drug that preferentially targets bacterial DHFR. Two independent constructs were prepared: 1) $SufT_{WT}$ or $SufT_{C62A}$ were expressed in fusion with C-terminus of F [3] fragment mDHFR and 2) Acn, SufR, SufS, SufU, SufT, and GCN4 fused to the C-terminus of F [1, 2] mDHFR fragments respectively. *Msm* as a host was transformed with plasmid constructs expressing $SufT_{[F3]}$/$GCN4_{[F1,2]}$ (negative control), $GCN4_{[F1,2]}$/$GCN4_{[F3]}$ (positive control), $SufT_{C62A[F3]}$/$SufT_{[F1,2]}$, $SufT_{[F3]}$/$SufT_{[F1,2]}$, $SufT_{C62A[F3]}$/$SufU_{[F1,2]}$, $SufT_{[F3]}$/$SufU_{[F1,2]}$, $SufT_{C62A[F3]}$/$SufS_{[F1,2]}$, $SufT_{[F3]}$/$SufS_{[F1,2]}$, $SufT_{[F3]}$/$Acn_{[F1,2]}$, $SufT_{C62A[F3]}$/$Acn_{[F1,2]}$, $SufT_{[F3]}$/$SufR_{[F1,2]}$, $SufT_{C62A[F3]}$/$SufR_{[F1,2]}$, $CFT10_{[F1,2]}$/$ESAT6_{[F3]}$ (positive control), and empty vector control expressing [F3] and [F1,2] only (negative controls).

## Purification of SufT and UV-vis spectroscopy

*Mtb sufT* (Rv1466) ORF was PCR-amplified using gene-specific oligonucleotides (Rv1466_Nde1_FP and Rv1466_Xho1; S1 Table), digested with *Nde*I-*Xho1*, and ligated into similarly digested His-tag-based expression vector, pET28a (TAKARA BIO, Clontech Laboratories, CA, USA) to generate pET28a-SufT construct. N-terminal histidine-tagged SufT was overexpressed in *E. coli* BL21 λDE3 by 0.6 mM IPTG [Isopropyl β-D-1-thiogalactopyranoside; MP Biomedicals, USA (16 h, 18˚C)]. To facilitate Fe–S cluster formation, cultures were incubated on ice for 18 min prior to induction and were supplemented with 300 μM ferric ammonium citrate and 75 μM L-methionine (Amresco, USA) and purified as described [16]. Single primer method of site directed mutagenesis approach was used to create cysteine to alanine substitutions (C62A). After the PCR, *Dpn*I was used to digest the wild-type methylated template plasmid. The reaction mixture containing the mutated *sufT* gene was used to transform *E. coli* BL21λDE3. Primer sequences are enlisted in S1 Table. Resulting clones were verified by sequencing, and the mutant $SufT_{C62A}$ variants of the $SufT_{WT}$ were purified as described earlier.

The UV–visible absorption spectroscopy of purified SufT protein was carried out in Thermo scientific spectrophotometer (Thermo scientific, USA) at 25˚C. Absorption spectra of $SufT_{WT}$ were recorded between λ 200 nm– 800 nm in sodium phosphate buffer (20 mM and pH 7.4).

## Iron estimation

For Fe estimation, freshly purified SufT was saturated with $FeSO_4$ (300 μM) and desalted using PD10 desalting column. Total Fe was quantified as described [71]. Total 100 μM and 0.1 mL of purified desalted SufT protein was heated at 95˚C for 30 min after treatment with 22% $HNO_3$ (0.1 mL). Samples were cooled to ambient temperature followed by addition of 0.6 mL ammonium acetate (7.5% w/v), 0.1 mL freshly prepared ascorbic acid (12.5% w/v) and 0.1 mL ferene (10 mM). The concentration of Fe present in the protein was determined by measuring the absorbance of the product, Fe-ferene complex at 593 nm, which was compared with a standard curve prepared from dilutions of freshly prepared Fe(III) solution in the range of 0–200 μM.

Total Fe was also estimated by atomic absorption spectroscopy (AAS) performed on atomic absorption spectrophotometer system (Shimadzu AA6880, Japan). AAS is a spectro-analytical procedure for the quantitative determination of ionizing elements using the absorption spectra. Total 200 μM of purified proteins of $SufT_{WT}$, $SufT_{C62A}$, BSA (negative control) and catalase (positive control) were saturated with 300 μM of $FeSO_4$ followed by desalting by PD10 column. Protein samples were estimate as described earlier [72] with little modifications. Briefly, equal volume of protein samples (100 μL) was cold digested with 10 mL of nitric acid overnight.

Next day 10 mL of di-acid (mixture of nitric acid and perchloric acid in the ratio of 9:3 ratios) was added and heat digested on sand hot plate. Digestion was confirmed by the sequential raising of red and white fumes. Precaution was taken to avoid charring of samples to get a final colorless solution. The digested samples were adjusted to 25 mL, using double distilled water and used for further iron analysis. Air compressor (4 psi) and gas (acetylene, 0.9 psi) were adjusted for smooth flame. Once AAS is ready, standard curve from 0, 1, 2, 3, 4 and 5 ppm of Fe (prepared from iron standard solution of 1000 mg/L of AAS grade, SRL) were generated followed by measurement of Fe in protein samples.

## Metabolite extraction and LC MS/MS based analysis

SufT–KD strain with and without ATc were grown in the standard 7H9 medium and harvested after 72 h of ATc treatment during the log phase ($OD_{600nm}$ of 0.8–1.4). Cells were quenched for 5 min in 4 volumes of 60% methanol (maintained at -45°C) and then centrifuged at 4800x$g$ (-5°C). The pellet was re-suspended in 700 μL of 60% methanol (maintained at -45°C) and then centrifuged at 4800x$g$ (-5°C). The pellet obtained was re-suspended in 1 mL of 75% ethanol and kept at 80°C for 3 min with intermittent vortexing at 1.5 min interval, immediately followed by incubation on ice for 5 min and centrifugation at 17000x$g$ for 15 min. The final supernatant was dried on a vacuum concentrator for 3–4 h and then stored at -80°C till further analysis. Metabolites were resolved and analyzed, with appropriate multiple reaction monitoring (MRM) methods, as described earlier [42]. At the time of metabolite measurement, the extract was dissolved in a suitable solvent as described [42] and injected into the Synergi Fusion-RP column on Agilent's 1290 infinity series UHPLC system coupled to a triple-quadrupole type mass spectrometer (Sciex QTRAP 6500).

## OCR and ECAR measurements

The SufT-KD strain grown for 72 h with or without ATc in standard 7H9 medium and equal number of cells ($2x10^6$ CFU/well) were adhered to the bottom of a XF cell culture microplate (Agilent technologies, USA), using Cell-Tak as a cell and tissue adhesive (Corning Cell-Tak). OCR and ECAR were measured using Agilent XF Extracellular Flux Analyzer (Agilent, US). Assays were carried out in unbuffered 7H9 media (pH 7.35) without glucose and ATc. Basal OCR and ECAR were measured for initial 21 min. Further, freshly prepared 2 mg/mL glucose as carbon source was added automatically in 7H9 unbuffered media, through port A and port B of the cartridge plate. To achieve maximum rate of respiration three measurements points were taken followed by 5 μM CCCP (Sigma-Aldrich, USA) treatment. Spare respiratory capacity, was calculated from % OCR value, by subtracting third basal reading, before glucose addition (normalized as 100%) from first point after CCCP addition.

## $H_2O_2$ and NO stress

Control and SufT-KD strains were grown in standard 7H9 media and treated with or without ATc for 72 h to deplete SufT. Cultures were diluted to an $OD_{600nm}$ of 0.15 and exposed to 0.5 mM, 1mM DETA-NO and 5mM and 10 mM $H_2O_2$. After 12 h cells were diluted appropriately and plated on ADS-7H11 plates. Colonies were counted after 3–4 weeks of incubation at 37°C.

## Cell line experiments

RAW264.7 murine macrophage cell line and PMA differentiated THP-1 cells were infected with Control strain with ATc and SufT-KD strains with and without ATc. Infection was performed at MOI 2 for 4 h, followed by washing thoroughly to remove extracellular bacteria with

warm PBS and suspended in the DMEM/RPMI-1640 media, containing 10% FBS. For CFU determination macrophages were lysed using 0.06% SDS and further serially diluted in 7H9 medium (0.05% Tween-80) and plated on OADC-7H11 at indicated time points. Colonies were counted after 3–4 weeks of incubation at 37˚C. For experiments with the iNOS inhibitor, 25 μM of 1400W was added to the media containing infected RAW264.7 after the 4 h of phagocytosis and remained throughout incubation.

## Animal experiments

For the chronic model of infection, 5–6 week old female BALB/c, mice (n = 6 per group) were infected by aerosol with approximately 200 bacilli per mouse with the SufT-KD strain using a Madison chamber by aerosol generation. Mice were divided in three groups, SufT-KD–DOX (without doxycycline), SufT-KD+DOX Acute phase (doxycycline started after 7 days of infection) and SufT-KD+DOX chronic phase (doxycycline started after 21 days of infection). Doxycycline was given in drinking water 1 mg/mL in 5% sucrose solution. Water bottles were light protected and replaced twice a week. At indicated times post infection, mice were euthanized, and the lungs were harvested for bacillary load, tissue histopathology analysis, and pathological scoring as described [46]. The remaining tissue samples from each mouse were homogenized and bacillary load was quantified by plating serial dilutions of tissue homogenates onto Middlebrook 7H11-OADC agar plates supplemented with lyophilized BBL MGIT PANTA antibiotic mixture (polymyxin B, amphotericin B, nalidixic acid, trimethoprim, and azlocillin, as supplied by BD; USA). Colonies were observed and counted after 4 weeks of incubation at 37˚C.

## Statistical analysis

All data were graphed and analyzed with Prism v8.0 (GraphPad) unless otherwise stated. Statistical analyses were performed using Student's t-tests (two-tailed). Where comparison of multiple groups was made either multiple t test or two-way ANOVA with multiple comparisons was performed. Differences with a $p$ value of $< 0.05$ were considered significant.

## Supporting information

**S1 Table. Details of primers used in this study.**
(DOCX)

**S1 Data. Excel spreadsheet containing, in separate sheets, the underlying numerical data and statistical analysis for Fig panels 3B, 4A-D, 5A, 5B, 6A-D, 7A-F, 8A, and 8D.**
(XLSX)

**S1 Fig. Modular structure and sequence alignment of DUF59.** (A) Nine modular structures of DUF59 containing proteins, referred to as S1 to S9. Sign (+) and (-) indicate corresponding SufT proteins that are within four ORFs of *sufBC* in the genome (associated) or encoded at distance of more than four ORFs in the genome (extraneous). N- and C-terminal motifs are indicated with different colors, along with their homologous functional role and host organism. The *Mtb* SufT (Rv1466) is a representative member of the S2 structure. (B) Alignment of the diverse amino acid sequences of DUF59 domains from the *H. sapiens*, *M. smegmatis*, *M. leprae*, *M. tuberculosis*, *S. meliloti*, *S. aureus*, and *B. anthracis* showing conserved putative site residues (DPE-X26–31-T-X2/3-C). The green box indicates extended N-terminal sequence exclusively found in *Mycobacterium* sp., yellow and red color indicates the locations of five α-helices and three β-strands, respectively. A highly conserved and putative hyper reactive cysteine (C62) is

star (*) marked.
(TIF)

**S2 Fig. UV- vis spectroscopy and total iron detection of SufT.** (A) Purified SufT was subjected to UV–visible spectroscopy. Absence of characteristic peak at 340 nm or 420 nm indicates that SufT is unlikely to be a Fe-S cluster containing protein. 100 μM of purified $SufT_{WT}$ and $SufT_{C62A}$ proteins were saturated with $Fe^{2+}$ in presence of $FeSO_4$ *in vitro* and total Fe was measured by B) biochemical assay and (C) atomic absorption spectrophotometer. Bovine serum albumin (BSA) and catalase were used as negative and positive control, respectively. (D) Western blot of $SufT_{WT[F3]}$ and $SufT_{C62A[F3]}$ expressed in host *Msm* for M-PFC experiments. Aconitase (Acn) is used as a loading control. Experiment was repeated three times and a representative image is shown.
(TIF)

**S3 Fig. Expression of *dCas9* only does not affect level of SufT.** (A) The *Mtb* strain expressing pRH2502/pRH2521 vectors as vector control (control) was exposed to ATc (200 ng/mL) for 24 and 48 h and the expression of *sufT* and *dCas9* was monitored by RT-qPCR. Experiment was performed in triplicate. (B) Growth curve in CFU/mL of control strain with and without ATc (200 ng/mL).
(TIF)

**S4 Fig. Depletion of SufT does not change ROS level.** Endogenous ROS was measured in the SufT-KD strain after 72 h of treatment with and without ATc (200 ng/mL) by using CellRox Deep Red dye staining. MFI indicates median fluorescence intensity. Cumene hydroperoxide (CHP; 5 mM) treatment for 15 min was used as positive control. Experiment was performed in triplicate with two independent experiments. ns: non-significant based on the Student's t-test.
(TIF)

**S5 Fig. SufT-KD is defective for growth in culture medium.** (A) CFU analysis of the SufT-KD strain at the indicated concentrations of ATc. Experiment was performed in triplicate with two independent experiments. Student's t-test was applied to measure significance ($p^{**} \leq 0.01$ and $p^{***} \leq 0.001$). **Defective survival of the** SufT-KD strain in RAW264.7 is partly dependent on iNOS. (B) iNOS inhibitor partially rescued growth defect of the SufT-KD in RAW264.7. RAW264.7 cells were infected with control+ATc, SufT-KD+ATc and SufT-KD-ATc at MOI 1:2, and further incubated with 25 μM of iNOS inhibitor 1400W (dotted lines) or without 1400W (solid lines). CFU was performed at the indicated time points. Two biological experiments were performed, and the student's T test was applied to calculate *p* value ($p^{*} \leq 0.05$, $p^{**} \leq 0.01$ and $p^{***} \leq 0.001$).
(TIF)

## Acknowledgments

We are thankful to BSL3 facilities at CIDR, IISc Bangalore for conducting all experiments related to *Mtb*. We thank Mass spectrometry facility at NCBS/inStem/CCAMP, Bangalore, for the access to, and use of mass spectrometers.

## Author Contributions

**Conceptualization:** Amit Singh.

**Data curation:** Chandrani Thakur, Nagasuma Chandra, Amit Singh.

**Formal analysis:** Ruchika Annie O'Niel, Sabarinath P. S, Chandrani Thakur, Raghunatha Reddy R. L., Nagasuma Chandra, Sunil Laxman, Amit Singh.

**Funding acquisition:** Amit Singh.

**Investigation:** Ashutosh Tripathi, Kushi Anand, Mayashree Das, Ruchika Annie O'Niel, Sabarinath P. S, Raghunatha Reddy R. L., Raju S. Rajmani, Nagasuma Chandra, Sunil Laxman, Amit Singh.

**Methodology:** Ashutosh Tripathi, Mayashree Das, Ruchika Annie O'Niel, Sabarinath P. S, Sunil Laxman.

**Project administration:** Amit Singh.

**Resources:** Amit Singh.

**Supervision:** Amit Singh.

**Validation:** Ashutosh Tripathi, Amit Singh.

**Visualization:** Ashutosh Tripathi, Amit Singh.

**Writing – original draft:** Ashutosh Tripathi, Amit Singh.

**Writing – review & editing:** Ashutosh Tripathi, Amit Singh.

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
