## [Decision Letter · Decision Letter 0]

13 Jan 2022

Dear Dr Singh,

Thank you very much for submitting your manuscript "Mycobacterium tuberculosis SufT integrates Fe-S cluster maturation with core intermediary metabolism for survival in vivo" for consideration at PLOS Pathogens. As with all papers reviewed by the journal, your manuscript was reviewed by members of the editorial board and by several independent reviewers. In light of the reviews (below this email), we would like to invite the resubmission of a significantly-revised version that takes into account the reviewers' comments.

Some additional work is required to support some of the claims in this study in particular analyzing the effect of SufT knockdown on viability since this could affect the interpretation of data in Fig. 6. The OD600nm measurement is not sensitive enough to detect changes in viability at the respective ATc concentrations and time points. It would be ideal to use inhibitors of iNOS and/or of ROS synthesis in the macrophages to demonstrate that the inability of the strain during SufT knockdown to grow in macrophages is due to increased sensitivity (at least in part) to these host defense systems. This would at least, in part, address the point raised by reviewer 1 that the attenuation in mice may simply reflect growth inhibition of the knockdown rather than killing due to host defenses.

We cannot make any decision about publication until we have seen the revised manuscript and your response to the reviewers' comments. Your revised manuscript is also likely to be sent to reviewers for further evaluation.

Sincerely,

Helena Ingrid Boshoff

Associate Editor

PLOS Pathogens

JoAnne Flynn

Section Editor

PLOS Pathogens

Kasturi Haldar

Editor-in-Chief

PLOS Pathogens

orcid.org/0000-0001-5065-158X

Michael Malim

Editor-in-Chief

PLOS Pathogens

orcid.org/0000-0002-7699-2064

Some additional work is required to support some of the claims in this study in particular analyzing the effect of SufT knockdown on viability since this could affect the interpretation of data in Fig. 6. The OD600nm measurement is not sensitive enough to detect changes in viability at the respective ATc concentrations and time points. It would be ideal to use inhibitors of iNOS and/or of ROS synthesis in the macrophages to demonstrate that the inability of the strain during SufT knockdown to grow in macrophages is due to increased sensitivity (at least in part) to these host defense systems. This would at least, in part, address the point raised by reviewer 1 that the attenuation in mice may simply reflect growth inhibition of the knockdown rather than killing due to host defenses.

Reviewer's Responses to Questions

**Part I - Summary**

Reviewer #1: This study characterizes the role of the M. tuberculosis (Mtb) iron-sulfur cluster biosynthesis enzyme SufT. SufT is the last gene in the suf operon, which contains several known Fe-S cluster biosynthesis genes. The role of SufT, however, is not well understood- there is no homology to other suf genes, except that it contains a DUF59 domain, which has also been described in S. aureus and yeast and other organsims as part of the Suf system. SufT interacts with other suf operon proteins, is essential under standard in vitro growth, is involved in maintaining TCA cycle intermediates, amino acids, glycolytic rate and respiration. SufT is also important for survival of Mtb in mice both during the acute and chronic phase.

This is a very thorough characterization of the Mtb SufT. The characterization of the domain of unknown function (DUF) in SufT is interesting, and the work goes all the way from biochemical analysis to in vivo testing in mice. Experiments are done well and controlled carefully. While DUF59 has been shown to be involved in Fe-S cluster biosynthesis in several other organisms before, and the novelty of the discovery of this protein’s function in Fe-S cluster synthesis is perhaps a little overemphasized, there is at least one idiosyncracy of the Mtb SufT in that it is also involved in protection and reactivation of Fe-S clusters. The interaction analysis by mycobacterial complementation assay is nice and shows interactions of SufT that had previously been missed. Metabolomics supports claims about the interactions of SufT with TCA cycle Fe-S cluster enzymes.

The novelty of the study is perhaps the weakest aspect of the study: A role of SufT in Fe-S biosynthesis was very likely, and some of the phenotypic analyses reveal fairly broad phenotypes expected when interfering with Fe-S clusters. The specific function of Mtb SufT that seems to distinguish it from other SufTs is repair, which adds a little bit of novelty. But the overall very thorough and technically sound characterization balances the somewhat incremental nature of the study.

Reviewer #2: In Tripathi et al. the authors characterize the function of the DUF59-containing essential protein Rv1466/SufT in Fe-S cluster biogenesis. The authors show that SufT interacts with other components of the Fe-S cluster biogenesis system, namely SufS and SufU. They alros report that SufT interacts with Fe-S cluster-containing proteins, such as aconitase and the suf regulator SufR, and sustains the activity of these proteins. TCA cycle, amino acid metabolism and sulfur metabolism are altered in a sufT-KD strain. Finally, SufT contribiutes to mycobacterial resistance to redox stress, intracellular survival in macrophages and lung colonization in the mouse model.

Overall, the manuscript is well written, and the conclusions rely on solid. The study as a whole brings important novel knowledge on sulfure metabolism in M. tuberculosis, and on the capacity of Mtb to cope with redox stress during infection.

Several aspects should be considered (see below).

Reviewer #3: This is an interesting manuscript exploring the role of the essential gene SufT in the iron-sulphur biogenesis machinery. The authors confirm that this gene is essential and perform some nice structural work on the protein demonstrating that it interacts with its partners SufS and Suf U as well as the central carbon metabolism enzyme aconitase. They go onto show that in accordance with this depletions of this gene alters the metabolites of central carbon metabolism, changes redox homeostasis and survival in both macrophage and murine of models of TB providing important infomation about the biological roles of these proteins. Its a significant body work which will of interest to TB researchers and those studying iron-sulphur proteins and the Suf system.

**Part II – Major Issues: Key Experiments Required for Acceptance**

Reviewer #1: No major issues

Reviewer #2: - The section "SufT depletion results in widespread metabolic realignment in Mtb" should be thoroughly reworded, and the claims should be toned down. Only correlations are reported here, not evidence of direct functional links. Similarly, the claim in the title of the manuscript should be mitigated. Metabolism is modified in the SufT-KD strain; the extent to which this is directly related to SofT activity is a complex question that would require far more experiments to be fully addressed.

- Fig. 4 would be more convincing if Fe-S clusters would be measured in Acn and SufR in the WT and sufT-KD strains.

- In Fig. 6 and 7, an important parameter to consider is the number/fraction of live bacteria in the sufT-KD population after 3 days of treatment with Atc. In 6A,B, to what extent if the reduced OCR and ECAR not due to reduced viability? Same in 7C, what is the fraction of live bacteria in the control (0 H2O2)?

- Claims lines 406-408 would be better supported by experiments using KO cells or inhibitors of NOS/ROS synthesis.

Reviewer #3: N/A

**Part III – Minor Issues: Editorial and Data Presentation Modifications**

Reviewer #1: Many of the quite broad phenotypes observed for SufT are not surprising and would be expected when disrupting Fe-S biosynthesis (derailed energetics, TCA imbalances). SufT being an essential protein complicates a rigorous and conclusive analysis of a depletion phenotype in mice. Although the authors do what they can to address this issue by using an inducible knockdown system, it is still difficult to know if the amount of doxycycline used in the mice leads to depletion to an extent that just arrests bacterial growth (as it does in vitro) rather than reveal a host-related phenotype. The paper is written clearly and Figures are also very clear.

Other comments:

Line 217: “(We next) asked if SufS directly associates with the proteins requiring Fe-S clusters…” I think this should read SufT.

Figure 1B: The side chains are difficult to see and the amino acid numbering is unclear: There is no C62 in the Mtb SufT sequence. If the numbering is following the B. anthracis numbering, including that sequence in the sequence alignment would help clarify. Also, do the other structures have Fe bound? Did the Mtb structure? That might explain the different distances of the highlighted residues.

Reviewer #2: - Fig. S2C (lines 169-172), the authors conclude that SufT does not contain Fe, catalase being used as a control (Fe-containing protein). This is not clear. It loks like there is Fe in SufT and that iron concentrations in SufT and catalase are similar. Please clarify.

- Replace SufS by SufT, line 217

- The conclusion, lines 250-252, should better be positioned within the next section.

- Line 343, Asp is not shown in the figure.

- In Fig. 5, why were S-containing amino acids not included? This should be discussed.

Fig. 8A, why is the WT strain not included?

Reviewer #3: This manuscript needs to be significantly edited as its a very hard read and the findings of the manuscript are really lost in all the details. The abstract needs to be completely rewritten and all the acronyms replaced so that the paper is attractive to readers and the key points are emphasised.

The in silico and other protein work whilst interesting needs to be significantly edited down and perhaps some of he details could be removed and put in the supplementary information. What is now shown about their biogenesis from this work needs to be clearly emphasised as its rather buried in places.

The metabolite measurements are interesting however this doesnt tell you anything about metabolic flux as inferred from the discussion. In order to do this the authors would have to perform some labelling.

PLOS authors have the option to publish the peer review history of their article (what does this mean?). If published, this will include your full peer review and any attached files.

Reviewer #1: No

Reviewer #2: No

Reviewer #3: No
---

## [Editor Report · Decision Letter 1]

25 Mar 2022

Dear Dr Singh,

We are pleased to inform you that your manuscript 'Mycobacterium tuberculosis requires SufT for Fe-S cluster maturation, metabolism, and survival in vivo' has been provisionally accepted for publication in PLOS Pathogens.

Best regards,

Helena Ingrid Boshoff

Associate Editor

PLOS Pathogens

JoAnne Flynn

Section Editor

PLOS Pathogens

Kasturi Haldar

Editor-in-Chief

PLOS Pathogens

orcid.org/0000-0001-5065-158X

Michael Malim

Editor-in-Chief

PLOS Pathogens

orcid.org/0000-0002-7699-2064

The authors have sufficiently addressed the reviewers' concerns.
---

## [Editor Report · Acceptance letter]

13 Apr 2022

Dear Dr Singh,

We are delighted to inform you that your manuscript, "Mycobacterium tuberculosis requires SufT for Fe-S cluster maturation, metabolism, and survival in vivo," has been formally accepted for publication in PLOS Pathogens.

Best regards,

Kasturi Haldar

Editor-in-Chief

PLOS Pathogens

orcid.org/0000-0001-5065-158X

Michael Malim

Editor-in-Chief

PLOS Pathogens

orcid.org/0000-0002-7699-2064